# Functional diversity of soil microbial communities increases with ecosystem development

Tord Ranheim Sveen [1] ✉, Maria Viketoft[1], Jan Bengtsson [1], Joachim Strengbom [1], Justine Lejoly [2], Franz Buegger[3], Karin Pritsch [3], Joachim Fritscher[4], Falk Hildebrand[4,5,6], Ernest Osburn [7], S. Emilia Hannula [2,8] & Mo Bahram [1,9,10]

Land abandonment is the single largest process of land-use change in the Global North driving succession and afforestation at continental scales, but assessing its impacts on soil microbial communities remains a challenge. Here, we establish a nationwide successional gradient of paired grassland and forest sites to track developments in microbial structure and functioning following land abandonment and gradual land-use change to forests. We show that microbes generally respond through threshold dynamics, leading to increasing functional but decreasing taxonomic diversity. Succession also entailed specialization of microbial nutrient (C-N-P) cycling genetic repertoires while decreasing genetic redundancy. This highlights a putative trade-off between two desirable ecosystem properties: functional diversity and functional redundancy. Fungal functional diversity underpins higher microbial C-cycling capacity, underscoring the link between functional traits and ecosystem processes. Changing litter quality similarly provides a mechanistic link between plant and microbial communities despite otherwise largely decoupled successional developments above- and belowground. While land abandonment is frequently touted as an opportunity to increase biodiversity and carbon storage, our results show that deeper knowledge about the multifaceted development of soil microbial communities and their links to plant communities during succession may be needed to fully grasp the impacts of global land abandonment processes.

Agricultural land abandonment (hereafter land abandonment) and afforestation are the two largest processes of land-use change in the Global North[1], profoundly altering landscapes, ecosystems, and biodiversity patterns[2]. It is estimated that ~500 Mha of agricultural land has been abandoned globally since the onset of industrialization in the mid-1800s[3]. After management ceases, abandoned fields are usually colonized by pioneer grasses and herbs, followed by shrubs, bushes and trees, until a stage of full afforestation is reached[4]. These changes

[1]Department of Ecology, Swedish University of Agricultural Sciences, Ulls väg 16, Uppsala, Sweden. [2]Department of Terrestrial Ecology, Netherlands Institute of Ecology, PO Box 50, 6700 AB, Wageningen, The Netherlands. [3]Research Unit for Environmental Simulation, Helmholtz Zentrum München, German Research Center for Environmental Health, Ingolstaedter Landstraße 1, D – 85764, Neuherberg, Germany. [4]Gut Health and Microbes Program, Quadram Institute, Norwich Research Park, Norwich, UK. [5]Decoding Biodiversity, Earlham Institute, Norwich Research Park, Norwich, UK. [6]University of East Anglia, School of Biological Science, Norwich, UK. [7]Department of Plant and Soil Sciences, University of Kentucky, Lexington, KY, USA. [8]Institute of Environmental Sciences, Leiden University, Leiden, The Netherlands. [9]Institute of Ecology and Earth Sciences, University of Tartu, Tartu, Estonia. [10]Department of Agroecology, Aarhus University, Slagelse, Denmark. ✉e-mail: tord.ranheim.sveen@slu.se

occur gradually, through a directional and largely predictable ecosystem development referred to as *secondary succession*[5] (hereafter succession). However, the traditional view of a putatively undisturbed "climax" forest (sensu Clements[6]) as the successional endpoint is increasingly disputed[7–9], and the vast majority of abandoned grasslands in Europe instead ending up as production forests incorporated into management regimes comprising low diversity, short rotation times, and clear-cutting[10–13]. The joint processes of land abandonment and forestry management pose serious threats to a range of organismal groups aboveground[14–18], but few studies have comprehensively examined their effects on soil microbial diversity and functioning.

Soil microbes constitute the largest pool of terrestrial diversity and undergo gradual changes in response to changing resource quality and soil properties during succession. For instance, soil fungal communities typically respond to decreasing litter quality of plants (resource-driven succession) through changes in the leaf-dry matter content (LDMC)[19], whereas bacteria are generally more sensitive to changes in abiotic properties like pH and soil organic matter quality (soil C:N)[20]. However, in contrast to the largely predictable patterns observed during plant succession, microbes often exhibit unimodal (i.e., hump-shaped) or threshold responses[21–23]. This could be related to sudden shifts in the metabolic trait distribution of the microbial decomposer communities[24], leading to abrupt transitions between microbial generalists and specialists as plant litter becomes increasingly recalcitrant to microbial degradation[25,26]. Ecological theory posits that niche differentiation and specialization of the microbial

saprotrophic community increase with increasing litter complexity and heterogeneity[27], and that this is accompanied by "tighter" and more efficient nutrient cycling processes. Yet, contrasting evidence also show high overlap of microbial genes related to nutrient cycling during succession[22,28], suggesting that metabolic traits remain unaltered during afforestation. This points to a high level of genetic redundancy, where the genes needed to perform nutrient cycling are widely shared across multiple microbial taxa[29]. Functional specialization and functional redundancy are both important features underpinning ecosystem functioning and stability, but few studies have reconciled these dimensions across ecosystems undergoing land-use change. Existing studies of microbial successional dynamics are moreover often hampered by limitations inherent to space-for-time designs (i.e., chronosequences)[30], which often fail to capture the spatial scales of land abandonment[13].

Here, we establish a nationwide gradient spanning from managed and open grasslands to abandoned and increasingly afforested sites undergoing succession (Fig. 1a, c). Each grassland site was paired with an adjacent forest site representing its land-use change endpoint[31] (Fig. 1b), typically a coniferous-dominated forest managed for wood production because historical land-use maps show that managed forest sites better represent the outcome of abandoned agricultural land than undisturbed forests[11,13,32–34]. We use amplicon (16S for bacteria, ITS for fungi) and metagenomic sequencing (Fig. 1d) to study how microbial taxonomic diversity and functional genes encoding enzymes involved in C-N-P cycling developed with abandonment and

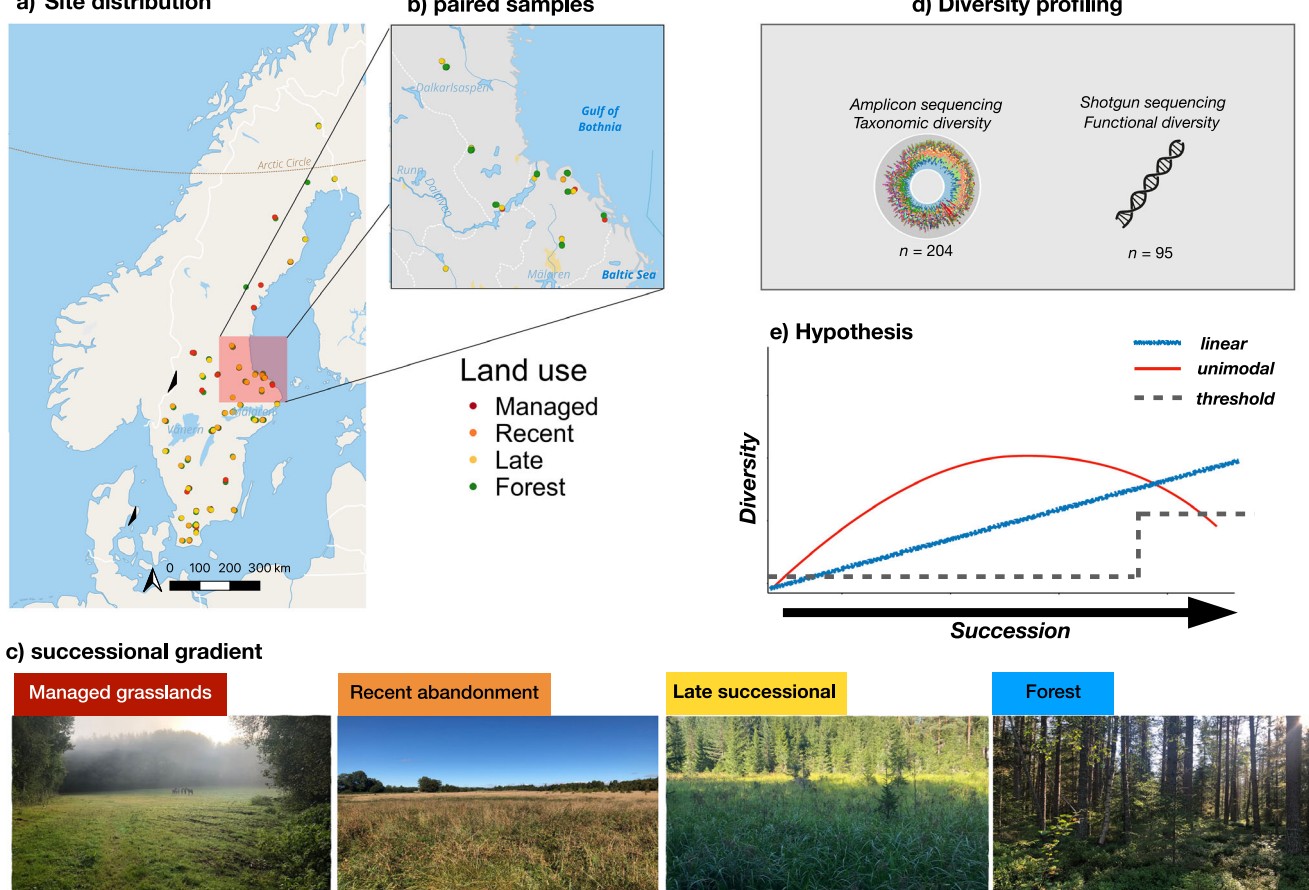

**Fig. 1 | Study design.** Distribution of (**a**) sites across a national successional gradient in Sweden, based on (**b**) a paired grassland and forest sites comprising (**c**) managed, recently abandoned, and late successional grasslands together with fully afforested reference sites representing the successional endpoint. Soil samples were gathered in the field and sequenced with (**d**) amplicon and shotgun metagenomics to yield taxonomic and functional microbial community profiles, with the hypothesis that these would change (**e**) linearly, unimodally, or with threshold responses to succession. Note that the trends are shown as positive, i.e., with increasing diversity here, but may also be negative.

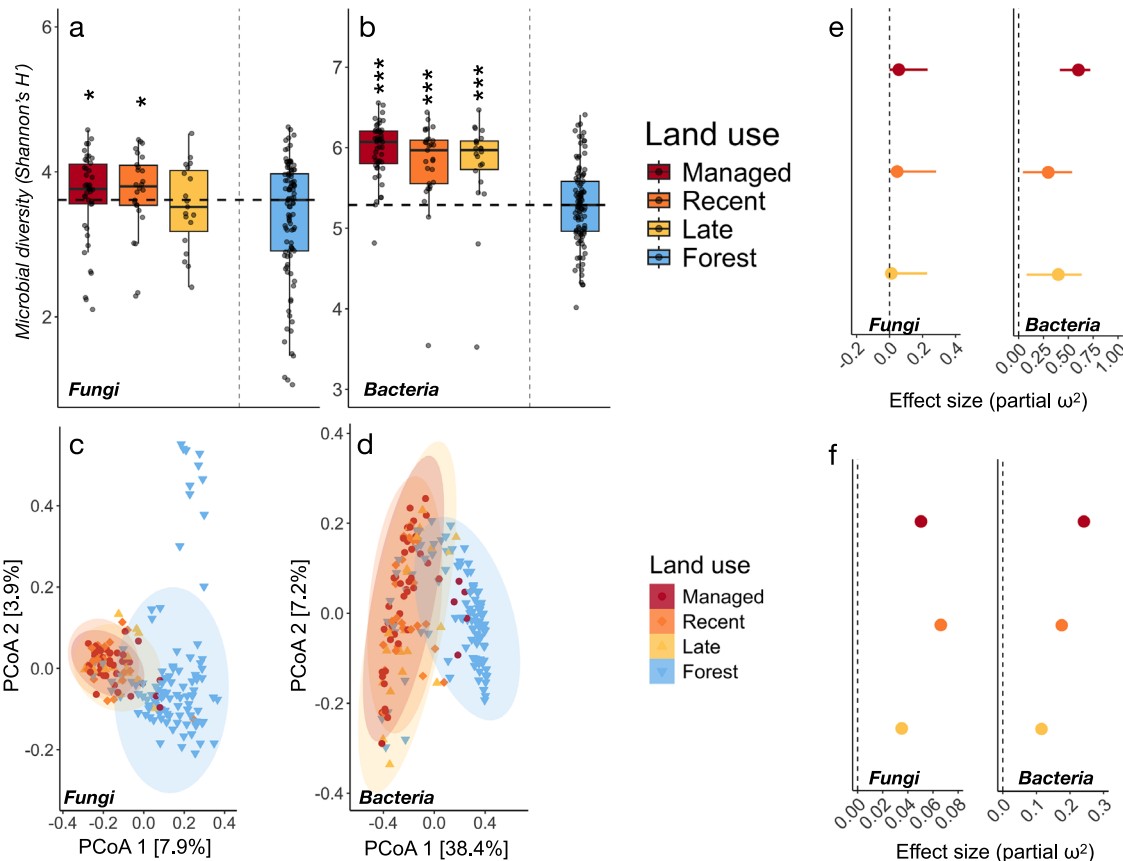

**Fig. 2 | Threshold changes to microbial diversity between grasslands and forests.** Taxonomic diversity (Shannon's $H'$) of (**a**) fungal ($n = 199$), and (**b**) bacterial ($n = 204$) communities across the land abandonment gradient. Boxes are bounded by the interquartile range (25th and 75th percentiles) of data, with the middle line showing the median. The box whiskers extend from the median to indicate minimum and maximum values at 1.5x the interquartile range. Points indicate the full distribution of sample values including minimum and maximum data values. The dashed horizontal line indicates the median of the forest sites, with asterisks (*) denoting significant differences between paired grassland and forest sites based on mixed-effect linear models accounting for paired structure and spatial distance with the following significance levels: $*p < 0.05$, $**p < 0.01$, $***p < 0.001$. Full test details are listed in Supplementary tables S6–S7. Principle coordinate analyses (PCoA) of (**c**) fungal, and (**d**) bacterial community composition during succession based on bray-curtis distances. Results from pairwise permutational multivariate tests (perMANOVA) for difference in community composition between land-use stages are found in Supplementary tables S8. Effect size estimates (partial $\omega^2$) for differences in (**e**) taxonomic diversity (i.e., alpha) and (**f**) community composition (i.e., beta diversity) between paired grassland and forest sites for each grassland stage, while including pair and distance between pairs as random factors. The dashed vertical line indicates no difference compared to forest sites and the lines in panel (**e**) indicate ±95 confidence intervals, with points denoting the estimated effect size. Data and code to reproduce this figure are available at: (https://zenodo.org/records/17176048).

afforestation, including links to measured degradation rates of a range of carbon substrates. We hypothesize that increasing resource complexity would increase the functional diversity and reduce the overlap of functional genes, leading to divergent levels of genetic redundancy and specialization with succession. We further expect genetic specialization and redundancy to be reflected in the degradation of simple and complex substrates, with high genetic redundancy leading to high degradation rates of low-complexity substrates and vice versa for specialization. Lastly, we expect these community changes to occur predominantly as unimodal or threshold-like responses (Fig. 1e).

## Results
### Threshold dynamics in soil microbial communities during grassland afforestation
Our successional gradient revealed a striking decrease of microbial (taxonomic) diversity and marked shifts in community composition with land-use change between grasslands and forests (Fig. 2). By contrast, grassland microbes were overall structurally and functionally similar, independent of the successional stage (i.e., managed, recently abandoned, and late-stage successional grasslands, Fig. 1c and of the increasing levels of afforestation (Table S6-7). The abrupt decrease of

bacterial diversity is consistent with threshold-like tipping points (Fig. 1e) occurring between late-stage successional grasslands and fully afforested sites (Fig. 2b). In contrast, fungal diversity declined gradually, and this evened out during the later stage of succession (Fig. 2a). Community composition differed strongly between grassland and forest sites (Fig. 2c, d), with fungal community composition additionally differed between managed and abandoned grasslands (Permanova: $p < 0.05$ for all comparisons, see Table S8 for full test details), indicating that a compositional shift occurs after the cessation of grassland management. Effect sizes (partial $\omega^2$) were derived from comparisons between paired grassland and forest sites while including pairs and spatial distance between pairs as random factors to account for the paired structure of the dataset. These showed overall larger effect sizes between managed grasslands and forests compared to abandoned and successional grasslands (Fig. 2e), and considerably higher for bacteria than for fungi (partial $\omega^2 = 0.46$ and 0.19 for bacterial alpha and beta diversity, and partial $\omega^2 = 0.07$ and 0.05 for fungal alpha diversity and beta diversity, respectively). Variable selection analyses revealed that thresholds in diversity changes coincided with a sharp decline in soil pH, increasing soil C:N ratio, and higher levels of leaf dry matter content (LDMC; Table S3–S4), whereas fungal diversity

was significantly related only to the geological parent material of the soils (Fig. S2a, c). Partitioning the contribution of these variables to the observed changes in alpha diversity revealed soil pH as the primary determinant of bacterial diversity decline during afforestation (Fig. S2d), whereas fungal diversity was mainly driven by changes in total soil organic carbon (SOC), parent material, and C:N (Fig. S2b). The inclusion of geographical distance between paired grassland-forest sites to account for spatial autocorrelation and climate factors (temperature, precipitation) did not alter these results (Table S6).

Because the paired forest reference sites were managed and heterogeneous in age and structure, we acknowledge that thresholds between late-stage successional sites and forests could be due to management factors rather than effects of afforestation. To test the influence of forest management on the results obtained here, we retrieved stand age and evenness as proxies of forestry management and examined whether the structure of microbial communities differed depending on these characteristics. Results showed that fungal community composition but not alpha diversity varied according to stand evenness (Permanova: Pseudo-$F_{1,2} = 1.28$, $p = 0.02$), whereas bacterial diversity and community composition did not vary with any of the factors (Table S9–S10).

### Functional diversity increases relative to taxonomic diversity during land abandonment

We next examined how the diversity of functional genes encoding enzymes involved in soil biogeochemical cycling (i.e., C, N, P-related genes) changed with succession and land-use change. Also, here, we found no differences between grasslands across differing stages of management and succession ($p > 0.05$ for all pairwise comparisons; see Table S11 for full test details). Conversely, threshold effects were notable in the response of fungal C-cycling genes, which increased in forests compared to grasslands (Fig. 3a–c). By contrast, the diversity of bacterial C-cycling and all microbial P-cycling genes remained at similar levels across all land uses (Fig. 3d–f). The diversity of bacterial N-cycling genes also decreased in forests compared to grasslands (Fig. 3g, h). As with taxonomic diversity patterns (Fig. 2e, f), effect sizes for functional genes were generally higher between managed grasslands and forests compared to successional (recent, late) grasslands (Fig. 3c, f, h). Forest age and evenness structure did not significantly affect the diversity and composition of any of the functional gene categories (Table S9, S10). Moreover, when juxtaposing the results obtained from the taxonomic and functional profiling (Figs. 2, and 3), a pattern of increasing functional relative to taxonomic diversity emerged, suggesting changes to microbial redundancy and functioning.

### Grassland abandonment leads to loss of genetic redundancy

The absence of direct relationships between taxonomic and functional diversity is considered a strong indicator of functional redundancy, as taxa can be replaced without losing genetic potential to carry out ecosystem functions[35,36]. To examine how the observed patterns in taxonomic and functional diversity related to functional redundancy, we combined all grassland sites into one single grassland category as the differences between successional stages were minimal (Figs. 2, and 3). Next, we used ordinary least-square regression (OLS) to test the strength and direction of the relationship between taxonomic diversity (predictor) and functional diversity (response variable). For fungi, the marked increase of C-cycling genes in the forest sites occurred despite a general loss of taxonomic diversity (Figs. 3a vs 2a), producing a negative relationship (OLS: df = 85, estimate = −0.15, $p = 0.011$; Fig. 4a) which was not significant in grasslands (OLS: df = 40, estimate = −0.04, $p = 0.694$). Comparing the slopes of both ecosystems (grasslands, forests) showed that these differed significantly (Anova: $F_{1,86} = 42.6$, $p < 0.001$), indicating that the coupling of taxonomic diversity and genetic C-cycling potential undergoes substantial alteration as

grasslands transition to forests. Similar results were obtained when assessing bacterial C-cycling and N-cycling genetic diversity (Fig. 4b, e). Both these genetic pools were moreover significantly related to taxonomic diversity in forests but not in grassland soils (OLS, $p < 0.05$ for both comparisons; see Table S13 for full test details). Because the number of functional genes scales with genome size for bacteria[37], the increase in functional genes relative to taxonomic diversity could be due to increasing bacterial genomes in forest soils. To account for this, we included average genome size (AGS) as a fixed effect in all models assessing bacteria, but this did not alter the direction or significance of the taxonomic-functional diversity relationships (Table S13). For P-cycling genes, diversity coupling was significant in both ecosystem types for bacteria (Fig. 4d), albeit stronger in forests than in grasslands (Anova, $F_{1,87} = 34.2$, $p < 0.001$). Interestingly, no corresponding coupling was found for fungi (Fig. 4c), indicating that functional redundancy may be relatively high when it comes to fungal P-cycling independent of ecosystem development. Soil pH and C:N were the main factor related to genetic C-cycling diversity for both fungi and bacteria (Fig. S3a–f), with hierarchical partitioning additionally showing high influence (~14% explained variation) of bulk density (Fig. S3b). The diversity of bacterial N-cycling genes was similarly driven by C:N but additionally showed high influence stemming from leaf litter quality (Fig. S3i, j). In contrast, the diversity of P-cycling genes was mainly driven by a mix of pH, total N content, parent material, leaf litter quality, and mean annual temperature (Fig. S3c–h).

### Widespread genetic specialization of the soil microbiome during ecosystem development

To examine whether microbial communities become more functionally specialized after land abandonment, we quantified the average niche overlap of C-N-P-related genes among successional stages (see "Methods"). Results showed decreasing genetic overlap of genes related to C- and P- cycling when going from managed grasslands to forests for both fungi and bacteria (Fig. 5a–d). Second-order polynomial regressions provided the best fits for P-cycling genes and bacterial C-cycling genes (Table S15), whereas fungal C-cycling genes decreased linearly with ecosystem development (Fig. 5a). In contrast, the overlap of N-cycling genes, calculated for bacteria only due to the lack of databases containing fungal N-cycling genes, showed a weakly unimodal pattern peaking in early successional grasslands (Fig. S4). We next partitioned genetic overlaps across differing substrates of increasing carbohydrate complexity (C-cycling; Fig. 5f, g) and pathways (N-P-cycling; Fig. S4, 5). Also here, we found that genetic overlap decreased across the full spectrum of simple and complex substrates. Notably, P-cycling specialization increased gradually for fungi, whereas the overlap between many of the P-cycling pathways decreased sharply between managed and abandoned grasslands for bacteria (Fig. S5).

### A diversity-redundancy framework for soil bacterial communities during succession

Our results strongly indicate that communities shift from high genetic redundancy to increasing genetic specialization during land-use change from grasslands to forests. We formalized and tested this putative shift using the framework developed in ref. 38, where functional and taxonomic dimensions of a community is combined to yield three interrelated properties describing taxonomic diversity, functional uniqueness (hereafter used synonymously with specialization) and functional redundancy. These interrelated properties can be further visualized along a main axis of redundancy and specialization (Fig. 6a)[39]. Importantly, as these analyses require traits-by-taxa matrices, we first predicted bacterial metagenomes from amplicon sequences ($n = 204$) using *picrust2*[40] and then inferred C-N-P-cycling genes based on KEGG orthology (see "Methods"). Predicted metagenomes are of limited value for analyzing specific pathways[41], so we

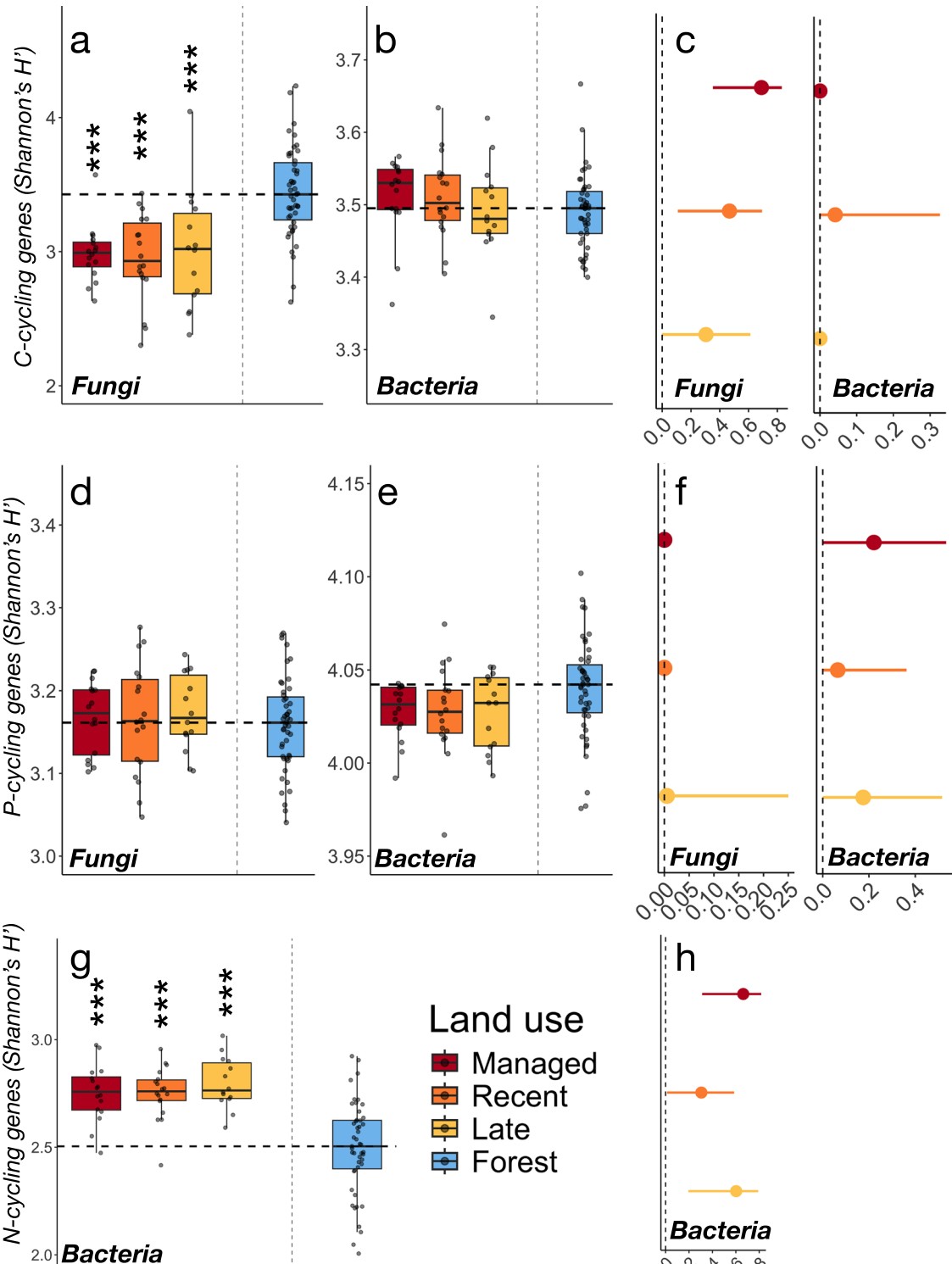

**Fig. 3 | Contrasting genetic diversity development during succession.** Genetic diversity of bacterial ($n$ = 95) and fungal ($n$ = 95) communities across the land abandonment gradient related to (**a, b**) C-cycling, (**d, e**) P-cycling and (**g**) N-cycling. Boxes are bounded by the interquartile range (25th and 75th percentiles) of data, with the middle line showing the median. The box whiskers extend from the median to indicate minimum and maximum values at 1.5x the interquartile range. Points indicate the full distribution of sample values including minimum and maximum data values. The dashed horizontal line indicates the median of the forest sites, with asterisks (*) denoting significant differences between paired grassland and forest sites based on mixed-effect linear models accounting for paired structure and spatial distance with the following significance levels: *$p$ < 0.05, **$p$ < 0.01, ***$p$ < 0.001. Full test details are listed in Supplementary tables S11–S12. Estimated effect sizes for differences in C-cycling (**c**), P-cycling (**f**), and N-cycling (**h**) between paired grassland and forest sites for each grassland stage, while including pair and distance between pairs as random factors. The dashed vertical line indicates no difference compared to forest sites, with points indicating partial $\omega^2$ effect sizes ±95 confidence intervals. Data and code to reproduce this figure are available at: (https://zenodo.org/records/17176048).

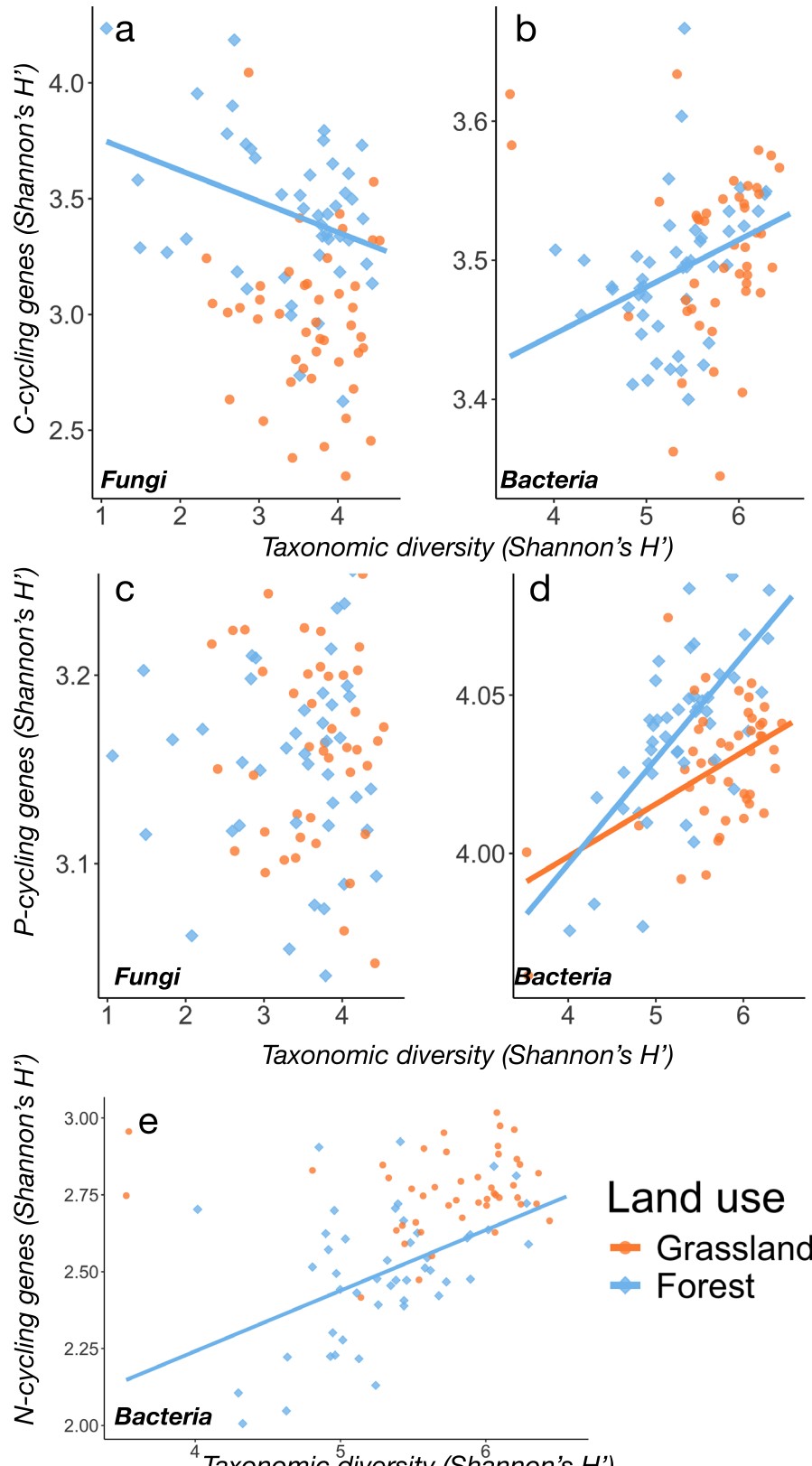

**Fig. 4 | Genetic redundancy decreases with change from grasslands to forests.** Scatter plots (**a**–**e**) showing the relationship between taxonomic and genetic diversity for fungal ($n = 95$) and bacterial ($n = 95$) communities. Solid lines indicate significant ($p < 0.05$) fit based on ordinary least-square regression, with full regression statistics listed in Supplementary table S13. Data and code to reproduce this figure are available at: (https://zenodo.org/records/17176048).

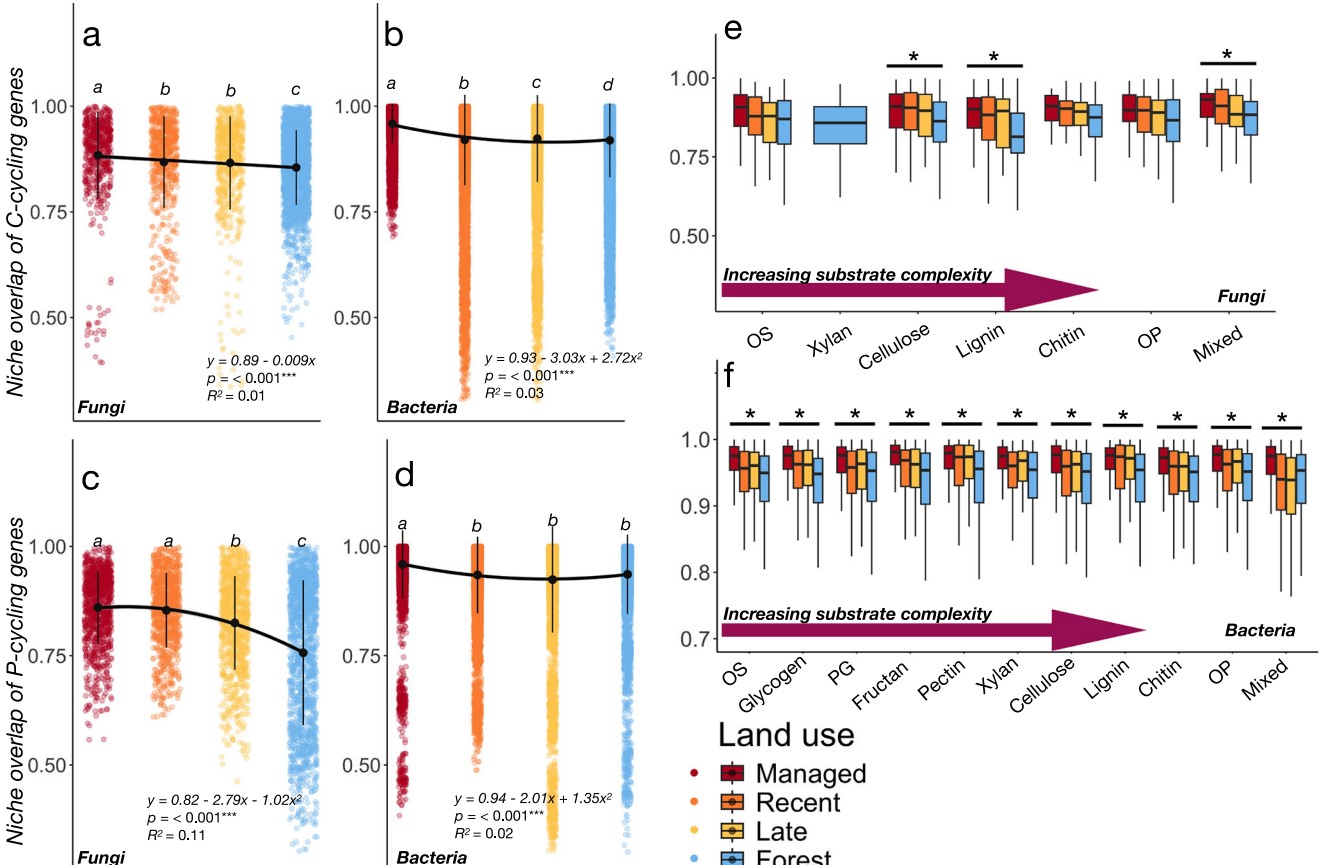

**Fig. 5 | Community-wide genetic specialization of microbial nutrient cycling increases with succession.** Niche overlap of carbon cycling genes for (**a**) fungal and (**b**) bacterial communities across the land abandonment gradient. **c**, **d** show corresponding niche overlap of P-cycling genes. An additional figure showing bacterial N-cycling genes is found in the Supplementary Materials Fig. S4. Black points show mean genetic overlap ± s.d. (vertical lines) for each land-use stage. Horizontal lines connecting land-use stages indicate significant ($p < 0.05$) ordinary least-square or second-order polynomial regression fits (Table S14). Letters indicate significant differences ($p < 0.05$) between land use-stages based on pairwise Wilcoxon Rank-Sum Tests (Table S15). In panels **e**, **f** the C-cycling have been partitioned according to their C substrate class and displayed for (**e**) fungi and (**f**) bacteria across a gradient of increasing substrate complexity. Sample size $n = 95$ for all plots (**a**–**f**). Boxes in boxplots are bounded by the interquartile range (25th and 75th percentiles) of data, with the middle line showing the median. The box whiskers extend from the median to indicate minimum and maximum values at 1.5x the interquartile range. Stars (*) above these show substrates where differences between one or more land-use stages are significant ($p < 0.05$) based on pairwise Wilcoxon tests, with full test results found in Table S16. Corresponding boxplots for P-cycling pathways are found in Supplementary Materials Fig. S5, with test results found in Table S17-S18. *OS* Oligosaccharides; *PG* Peptidoglycan; *OP* Other polysaccharides; Mixed = Mix of oligo- and polysacchcarides Data and code to reproduce this figure are available at: (https://zenodo.org/records/17176048).

focused only on overall changes in the diversity of metabolic pathways between ecosystems. Based on the predicted metagenomes, we assessed redundancy and specialization under the hypothesis that grasslands and forest sites would cluster differentially along the vertical axis separating these diversity dimensions (Fig. 6a). As expected, functional redundancy related to C-cycling decreased sharply between grasslands and forests, whereas functional specialization instead increased (Fig. S6), consistent with a threshold response to afforestation occurring late in the successional gradient. The quantity (Total C) and quality (C:N) of organic carbon were the main variables driving redundancy and specialization (Fig. S7a), differing from the factors affecting bacterial alpha diversity and community composition (Fig. S2). Clustering of grassland and forest sites occurred mainly along the redundancy-specialization axis, in line with our hypothesis (Fig. 6b; Fig. S8), and these results were further corroborated by compositional differences between the two land-use types (Table S19). Notably, N- and P-cycling showed the same redundancy-specialization divergence (Fig. S8), whereas no differences were found between any of the differing grassland successional stages. However, the main factors underlying these results varied, with N-cycling more related to soil pH in addition to soil carbon quality (Fig. S7b) whereas P-cycling was related to the successional stage of the site and plant leaf traits (Fig. S7c).

## Functional diversity is of key importance for substrate degradation

Lastly, we tested whether increasing functional diversity and specialization during succession would relate to degradation rates of substrates across a gradient of increasing complexity. For this, we incubated a subset of paired grassland-forest samples ($n = 154$) and measured substrate-induced respiration (SIR) across six substrates ranging from easily degradable oligosaccharides (glucose) to complex recalcitrant substrates like lignin and chitin. Degradation rates were generally higher in forest than in grassland soils but increased with abandonment and succession (Fig. 7a, and Fig. S9). Interestingly, the least and the most recalcitrant substrates (glucose, chitin) were exceptions to this, with no differences between any of the grassland and forest soils (Table S19). When combining the full range of degradation rates into an aggregate measure of functional capacity (MSIR, see "Methods") and using this as a response variable with microbial taxonomic and functional diversity as predictors, we found that MSIR was most associated with fungal C-cycling genetic diversity (Fig. 7b).

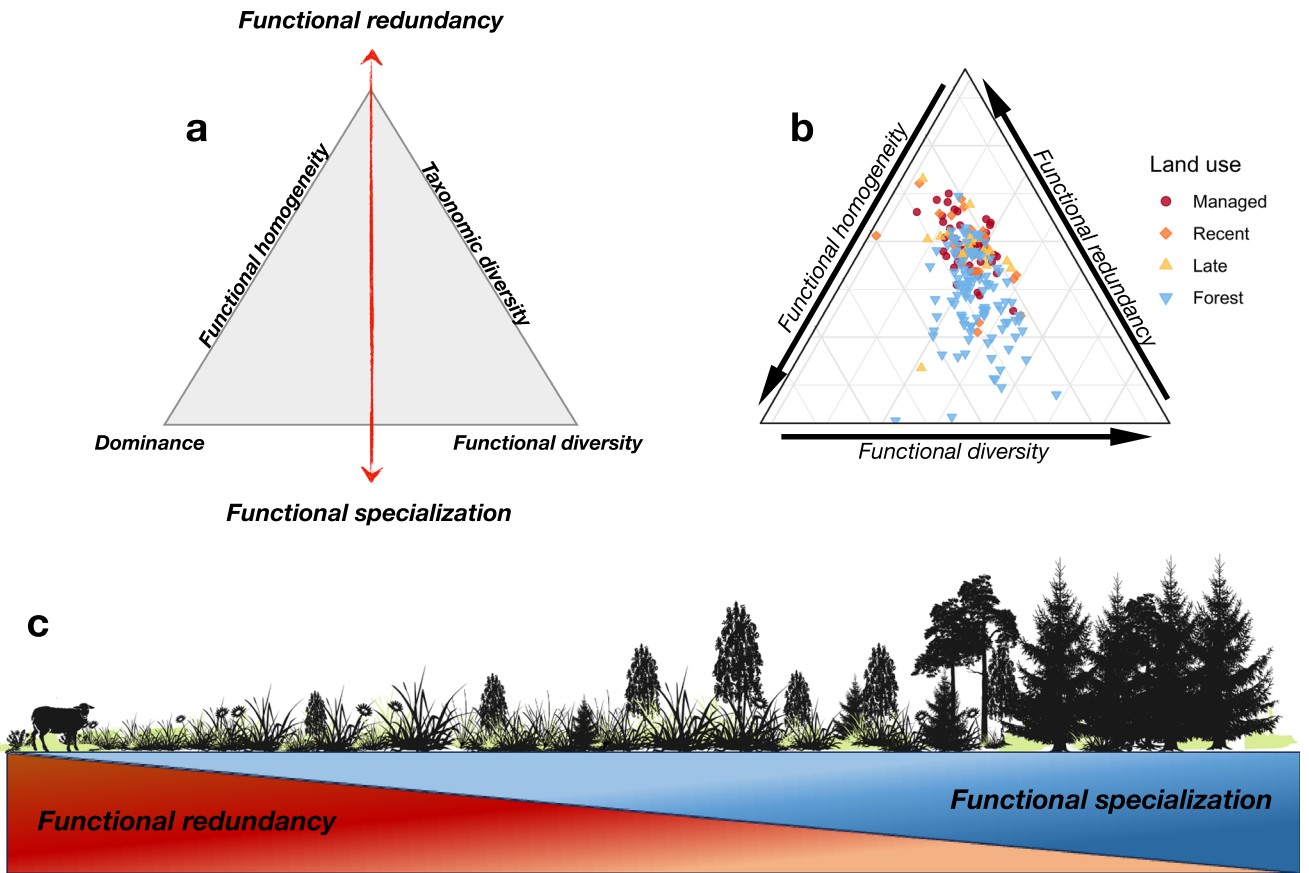

**Fig. 6 | A general framework of redundancy-diversity divergence during succession.** Conceptual ternary plots showing (**a**) interrelated components of taxonomic and functional diversity and how these can be arranged conceptually to produce an axis of functional redundancy-functional specialization. Ternary plot (**b**) showing the placement of grassland and forest sites according to the diversity metrics shown in (**a**) of bacterial communities inferred from predicted metagenomes (*n* = 204). Corresponding ternary plots for P- and N-cycling can be found in the Supplementary Materials Fig. S8. See refs. 30,31 for a mathematical and conceptual elucidation of the interrelated diversity components shown in (**a**). Conceptual gradient (**c**) depicting the redundancy-specialization axis during succession after land abandonment. Data and code to reproduce this figure are available at: (https://zenodo.org/records/17176048).

Notably, this relationship did not differ between grasslands and forests (Anova: $F_{1,70}$ = 0.015, *p* = 0.903), suggesting fungal control over substrate degradation across the whole range of grasslands and forests investigated. In contrast, we found no corresponding relationship for bacterial C-cycling genetic diversity (Fig. 7c). MSIR moreover correlated negatively with bacterial taxonomic diversity (df = 143, *r* = −0.37, *p* < 0.001) and showed no significant relationship with fungal diversity (df = 142, *r* = −0.06, *p* = 0.49).

## Discussion

Elucidating the patterns and processes occurring after land abandonment is a notorious challenge in ecology because the gradients used to infer ecosystem development are spatially confined and fail to match the extent of land abandonment. By establishing a nationwide gradient of paired grassland and forest sites (Fig. 1), we could examine how microbial communities respond to land abandonment and land-use change at an unprecedented scale. Contrary to the notion of the putatively undisturbed "climax" forests (sensu Clements[6]) inherent to much successional theory, we opted for using conventionally managed forest sites across a broad range of stand age and evenness (Fig. S1) as reference endpoints for adjacent grasslands. This choice reflects an assumption that grasslands, once abandoned and if not restored, will predominantly end up as managed production forests, as reflected in historical land-use maps across Sweden and Europe[11,13,32–34].

We found that microbial community responses during grassland abandonment and afforestation varied depending on organismal group (i.e., bacteria vs fungi) and the diversity aspect examined (i.e., taxonomic vs functional diversity). For bacteria, considerable taxonomic diversity was lost in the transition between late-stage successional grasslands and forests (Fig. 2b), whereas the corresponding decrease of fungal diversity was gradual (Fig. 2a). The overlap of annotated C, N, and P-cycling genes similarly decreased across the land-use gradient (Fig. 4a-d), despite overall few changes (Fig. 3b–d) or threshold dynamics (Fig. 3a, e) of the overall functional gene diversity. These results both confirm and contrast with our hypothesis and with foundational successional theory, where development is thought to occur as a gradual and largely predictable series of changes, and highlight discrepancies across the community properties examined. Reduced overlaps of annotated functional genes were generally better fitted to second-order polynomial than OLS-regressions (Table S14), indicating that changing niche space is non-linear and likely driven by a mix of the change in resource quality with changing plant community composition and by abiotic filtering[22]. Metabolic specialization of the soil microbiome occurring during grassland succession is therefore likely to be reflected in a non-linear transition between generalist and specialist taxa, consistent with findings within the overall framework of the fundamental microbial metabolic niche[42] where non-linear taxa overlap and transitions between generalist and specialist have been previously shown across environmental gradients[43,44]. Indeed, a recent survey of more than 230 soil bacterial communities across a continental scale showed similar bimodal distributions of generalist and specialist taxa, based on their average niche breadth[26].

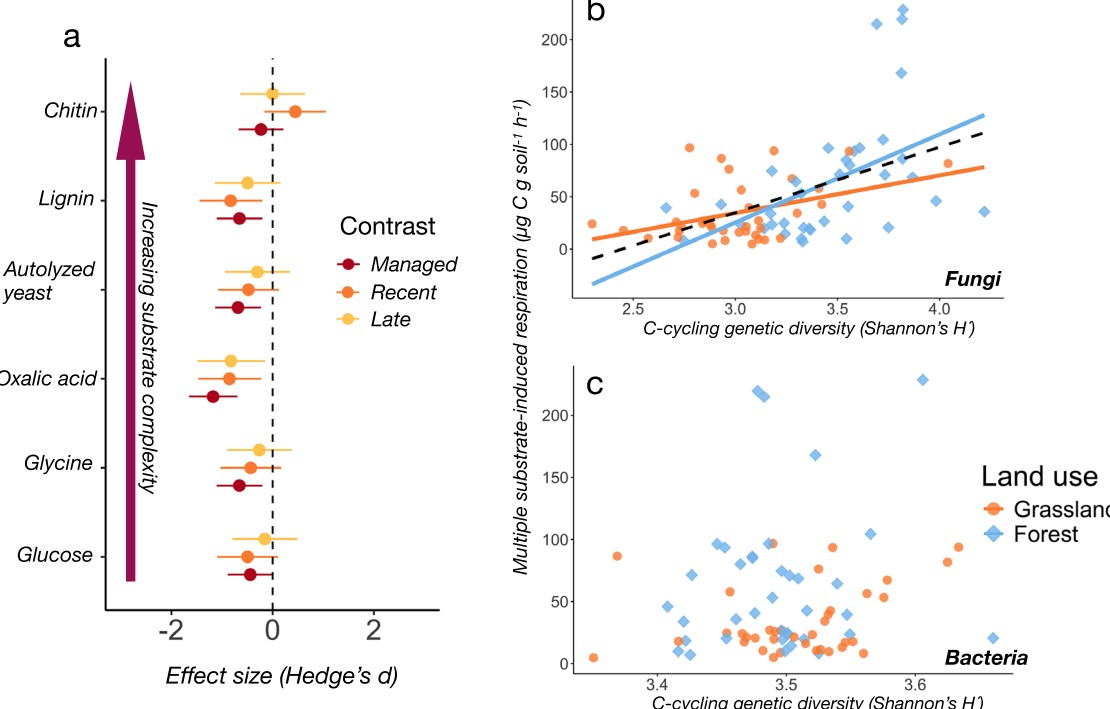

**Fig. 7 | Fungal functional diversity drives substrate degradation in grassland and forest soils. a** Effect sizes of substrate-induced respiration (SIR) rates between paired grassland and forest sites for managed, recently abandoned and late successional grasslands for a range of substrates of increasing complexity starting from glucose and ending in chitin. Points to the left of the dashed line indicate higher respiration rates in forests than in grasslands, whereas points to the right of the dashed line indicate higher grassland respiration rates than forest respiration rates. Points denote estimated effect sizes ± 95 % confidence intervals. Scatter plots showing the relationship between (**b**) fungal, and (**c**) bacterial genetic C-cycling diversity and the aggregate respiration rate of all substrates (MSIR). Solid lines indicate significant (two-tailed F-test, $p < 0.05$) fit of the regression line for each land-use type (grassland, forest). The dashed line in (**b**) shows the overall fit of all points independent of land-use type. Sample size $n = 154$ for plots (**a**–**c**). Data and code to reproduce this figure are available at: (https://zenodo.org/records/17176048).

Here, we observed that threshold responses in taxonomic and functional diversity coincided with a sharp decline in soil pH, increasing soil C:N, and higher levels of leaf dry matter content between grassland and forest sites (Table S3). These are all emblematic changes in soils undergoing succession after land abandonment in boreal biomes[19]. Of the key ecosystem properties identified, soil pH, C:N, total soil organic carbon (SOC), and leaf quality (LDMC) were moreover consistently singled out as key properties related to both taxonomic and functional diversity in both variable selection analyses and in hierarchical partitioning of fixed effects (Fig. S2, S3). Notably, LDMC provides a powerful "functional marker" of multiple ecosystem processes during secondary succession[45], including microbially driven litter decomposition[19], whereas increases in SOC have been linked to increasing microbial taxonomic diversity and enzyme potential during secondary succession previously[22]. The depletion of base cations through tree growth leading to soil acidification[46] and a build-up of recalcitrant leaf litter[45] could be important processes underpinning these abiotically driven changes, whereas the accumulation of plant detritus during afforestation, visible in the increasing levels of SOC in forest sites (Table S3), has also been shown to support a more active microbial community[47]. This effect is however likely modulated also by the quality of the litter[22]. Thus, together, LDMC, C:N, and SOC reflect changing resource quality and input during grassland afforestation which, together with soil pH, profoundly alter soil microbial community structure and functioning.

Afforestation of grasslands increase soil acidity by several pH units[48], but to our knowledge, there are no time-series to indicate whether this development tends to be linear or threshold-based. Because our chosen forest sites were generally managed, the observed threshold effects could also reflect a "management gap" between late-stage successional grasslands and the fully afforested endpoints, potentially setting these apart from the trajectory of a naturally developing ecosystem after abandonment. Yet, as we found generally few differences in the taxonomic and functional components of the soil microbiome across forests of differing ages and evenness classes (Table S9-10), we are inclined to think that forest management is not a main driver of the thresholds observed. A similar gap between late-stage successional grasslands and an unmanaged forest reference site was moreover found in ref. 22, indicating that threshold dynamics late in the successional gradient could be an intrinsic feature of microbial community development after land abandonment. Resolving this question likely requires time-series that circumvent the pitfalls of space-for-time substitutions altogether[30].

The absence of significant correlations between taxonomic and functional diversity is indicative of high functional redundancy, because removing taxa in redundant systems will thereby not affect the genetic diversity[49]. In other words, functional redundancy is visible in the decoupling between taxonomic and functional diversity[50]. We found differences in this relationship between grasslands and forests, with notably higher functional diversity relative to taxonomic diversity in forest soils (Figs. 2–4). This suggests that fewer taxa contribute to the overall pool of genetic diversity in forest soils and is corroborated by the generally strong associations between taxonomic and functional diversity in these sites but not in grassland (Table S13). These relationships were independent of the communities' average genome size (AGS; Table S13), suggesting that a shift toward taxa with larger genome size is not the main driver behind the loss of bacterial redundancy in forest soils[51]. By contrast, larger fungal genomes in forest soils could explain the negative relationship observed between fungal taxonomic and C-cycling genetic diversity (Fig. 4a), where taxa

with smaller genomes containing fewer C-cycling genes are presumably replaced by fewer taxa with larger genomes harboring more genetic diversity. If so, this would be another indication of functional specialization, in line with the decreasing overlap of fungal C-cycling genes across the gradient (Fig. 5a). Fungal genome sizes have been reported to increase in nutrient-poor soils[52], where fertility was defined as the availability of inorganic N and P. Given the large increase of SOC and C:N in forest compared to grassland soils here, it is plausible that the same elements are stoichiometrically limited and could yield the same pattern of increasing fungal genomes. However, due to the lack of tools to reliably infer AGS for eukaryotes, this could not be tested explicitly here. We also note the generally negligible influence of climate (mean annual temperature and precipitation) on microbial alpha and beta diversity (Figs. S2, S3), indicating consistent microbial successional dynamics across differing climatic zones.

The generally decreasing overlaps in the genetic repertoires of both fungal and bacterial communities with succession moreover indicate that specialization occurrs in all measured aspects of nutrient cycling (Fig. 5a–d; Fig. S4) and across all pathways (Fig. 5e, f, Figs. S4, 5). This is in line with recent evidence showing that bacterial specialization along one axis of environmental variation may explain up to 80% of the variation in all other axes[26], meaning that specialization within only one of multiple metabolic dimensions (e.g., P-cycling) is rare. Yet, it is important to acknowledge that the lack of databases for functional annotation of eukaryote N-cycling genes impedes similar analyses of fungal N-cycling gene redundancy and specialization.

Whether the loss of genetic redundancy also translates into a loss of functional redundancy is an open question[50]. Conversely, we may ask whether increased genetic diversity and specialization increase the rates or efficiency of ecological processes such as C-N-P-cycling. We measured the degradation rates of a range of increasingly complex substrates representative of changing litter quality during succession and found that these were generally higher in forest than grassland soils (Fig. 7a). This contrasts with our hypothesis of higher degradation rates in grassland due to genetic redundancy of C-cycling genes and suggests that functional specialization may be an important factor driving degradation of individual litter chemicals during decomposition[53], possibly due to higher complementarity in communities with low metabolic overlaps. In forest soils, the litter-derived pool of organic matter is typically dominated by structurally complex substrates enriched in recalcitrant lignin and chitin[54], and the enzymatic degradation of these into progressively simpler molecules selects for a highly specialized web of microbial cross-feeding[55]. By contrast, grassland soils are characterized by simpler and more easily degradable carbohydrate inputs, in which metabolic generalists with broad enzymatic repertoires dominate the decomposer community[56]. Thus, efficient resource partitioning among a community of specialist taxa provides a framework through which the higher substrate degradation rates in lower-diversity forest soils can be plausibly explained. This is in line also with a recent study finding both empirical and theoretical evidence for increasing dominance of specialist bacterial taxa with the supply of additional glycolytic carbon sources[57], which is likely to be an accurate reflection of the build-up of multiple sources of increasingly complex substrates in forest soils during succession[58]. However, we stress the observational nature of our study and that that our results are not necessarily causative, underscoring the need for more research to trace carbon flows throughout the decomposer community and pinpoint its mediating metabolic pathways. We also observed significant correlations between fungal but not bacterial diversity of C-cycling genes and aggregated degradation rates (i.e., MSIR) across all substrates in both grassland and forest soils (Fig. 6b), suggesting that saprotrophic fungi are of high functional importance also in these systems.

Functional diversity and functional redundancy are both crucial components supporting ecosystem functioning and stability[59–61]. Our results indicate that these two diversity components may be inversely related during succession (Fig. 6c), in line with our first hypothesis. Using computationally predicted bacterial metagenomes, we found that grassland and forest sites clustered regularly along the redundancy-specialization axis for all nutrient cycling processes (Fig. 6a; and Fig. S8), and that this clustering was likely driven by changing soil C:N, pH, and LDMC (Fig. S7). This corroborates our hypothesis that changes in resource complexity and soil acidity trigger shifts from taxonomically diverse but functionally redundant (i.e., more similar with respect to functioning) to functionally rich communities with more specialized communities. An intriguing outcome of this presumed trade-off is that ecosystems undergoing afforestation after land abandonment may become more efficient when it comes to carbon- and nutrient cycling due to higher functional diversity and specialization, but less resilient to disturbances due to lower functional redundancy[60]. Again, we emphasize that these results are based on the diversity of metabolic traits, here inferred through functional annotation of genes at community level, for which the relationship with actual process rates is highly context dependent[62,63] and may blur at the level of higher-order functions such as decomposition[50]. Similarly, the divergence pattern identified here relates to predicted metagenomes of soil bacteria only, with the absence of adequate tools to predict fungal metagenomes from marker sequences precluding corresponding analyses for fungi and will need to be verified using communities with known taxonomy and functional profiles. However, while the idea of a redundancy-specialization trade-off has been previously proposed in the context of global bird, mammal, and plant distributions[39,64], it has not yet been evaluated and tested for soil microbes. Our results provide a valuable insight into these dynamics in the context of the well-studied phenomenon of ecological succession, anchored in a nationwide gradient with high generalization. Based on these findings, we predict that other findings of negative relations between taxonomic and functional diversity across diverse environmental gradients[51,65,66] would yield the corresponding redundancy-specialization trade-off observed here. In conclusion, our study reveals that grassland abandonment and afforestation entail a broad range of changes to the soil microbiome when compared to adjacent forest sites, including a loss of taxonomic diversity while the diversity of functional genes encoding for C-N-P-related enzymes remains stable or increases. These dynamics imply threshold responses and a gradual transition from high genetic redundancy to functionally specialized microbiomes, with potential consequences for their capacity to degrade differing sources of carbohydrates. Further studies are needed to investigate whether the transition from functionally redundant to specialized microbiomes affects the long-term stability of ecosystems.

## Methods
### Study design
The study comprised 102 grassland and 105 forest sites (Table S1) derived from environmental monitoring programs conducted at the Swedish University of Agricultural Sciences (SLU). Grasslands were initially categorized into differing successional stages (Managed, recently abandoned, Late successional) based on site descriptions, and this was corroborated by permutational distance-based multivariate analyses (db-MANOVA, Table S2) for differences in plant community composition coupled with indicator species analysis[67]. As historical records show that abandoned grasslands in Sweden predominantly end up as conventionally managed production forests[11–13,68], we selected production forests (i.e., > 1 m³ ha⁻¹ year⁻¹ biomass production) as the reference endpoint for the abandoned grasslands. These forest sites were characterized by considerable variation in stand age and evenness (Fig. S1), indicating differences in management and/or growth cycle. To increase the accuracy of the forest reference, we paired each grassland with an adjacent forest site based on a

geographical proximity criterion (< 6.5 km distance between paired sites, median distance = 3.63 km), under the explicit assumption that grasslands, which, if and when abandoned and fully afforested, would share approximately the same vegetation and soil microbiome characteristics as their adjacent forest sites[30]. Altogether, 190 (92 %) of the 207 sites were paired. Although geographical pairing ensures that sites are located within the same regions and increase the likelihood that grassland and forest sites share broadly the same environmental characteristics, we acknowledge that local differences in climate, parent material, and topography exist at this spatial scale and could lead to discrepancies in the expected successional trajectory. Climate data comprising mean annual temperature (MAT) and precipitation (MAP) for each site was collected from the Swedish Meteorological and Hydrological Institute's (SMHI) long-term local weather stations. Information about the parent material for each site was gathered from the Swedish Geological Survey (SGU).

## Soil sampling

The sites were sampled between July and September 2020 following the standardized protocol established in ref. [69]. Briefly, at each site, 15 samples were collected across a circular area of ~ 200 m$^2$ using a soil corer (diameter = 3 cm, depth = 10 cm), and pooled into a bulk sample. Surface layers constituting loose litter and fibric material was removed prior to sampling but otherwise no distinction was made between the organic and mineral layers when sampling. By sampling at a standard soil depth, we therefore compare samples with differing levels of organic horizon depths, which strongly shapes resource availability, mineralogy, and physicochemical conditions. In particular, peaty soils with no discernible mineral horizon differ strongly from mineral soils. A total of 12 forest sites (11 % of all forest soils) were peaty soils without any mineral horizon, but this category also comprised a total of 7 grassland soils (6.8 % of all grassland sites). The alternative approach to sample at a consistent 0-10 cm depth in the mineral horizon was discarded as this would have failed to capture most of the resource-driven community changes occurring in the organic soil horizon. A subset of the pooled sample was air dried (< 40 °C) within 24 h of collection and stored in a zip-lock plastic bag with silica gel to minimize humidity and prevent development of molds during transit and stored frozen (−20 °C) until molecular analyses. All equipment was sterilized with 95% ethanol between sites.

## Abiotic drivers of soil diversity

Approximately 15 g of soil was used to analyze pH (1:5 soil:water suspension). Available phosphorus (P-AL) and potassium (K-AL) were extracted using ammonium lactate and acetic acid at pH 3.75[70] and analyzed using the stannous chloride-molybdate procedure (P-AL) and inductively coupled plasma atomic emission spectroscopy (ICP-AES), respectively. Exchangeable calcium and magnesium concentrations were measured in ammonium acetate extract (pH = 7.0) using ICP-AES at Agrilab Uppsala, Sweden. Total C and N contents was determined on aliquots (1-20 mg) of air-dried soil using an Elemental Analyzer (Euroa EA, Eurovector, Milano, Italy). We used leaf dry matter content (LDMC) as a proxy for leaf quality, as this has been shown to correlate well with litter quality and decomposition in successional systems in our study area[19] LDMC values were derived from the TRY database[71] (Ver. 6.0) and covered 120 out of the total 235 plant species (51.1 %) recorded across all sites, and converted to community-weighted means. While we acknowledge that traits derived from plant databases disregard intraspecific variation that could affect the results, it has been shown that interspecific trait variation is of considerably greater importance than intraspecific variation[72].

## Substrate-induced respiration

We performed catabolic profiling on a subset (n = 156, or n = 77 paired grassland-forest sites, Table S1) of air-dried soils using substrate-induced respiration (SIR) for a range of carbon substrates of differing complexity, following the protocol established in refs. [73,74]. Briefly, 8 mL solutions of glucose, glycine, oxalic acid, autolyzed yeast, lignin, and chitin was added to 4 g dry-weight equivalent of fresh soil (1 analytical replicate per solution). After 1 h of pre-incubation, soils were incubated for 4 h at 20 °C except for lignin and chitin which were incubated for 24 h. After incubation, respiration for each amendment was determined using Gas Chromotography (Trace CG Ultra Gas Chromatograph (Thermo Fisher Scientific, Milan, Italy)). Basal respiration rates, measured using 8 mL of added ddH$_2$O, were subtracted from each substrate-induced respiration rate to account for the added effect of water flushing (Birch effect)[75]. We used Multiple substrate-induced respiration (MSIR) as a proxy for the communities' catabolic profile and total functional capacity[76], which was calculated by subtracting basal respiration rates from each substrate-amended soil, and subsequently summing the individual substrate respiration rates for each sample.

## Molecular analyses

**DNA extraction and sequencing.** DNA was extracted from 200 mg of dried and milled soil samples using the PowerMax Soil DNA Isolation Mini kit (Qiagen GmbH, Hilden, Germany) following the manufacturer's instructions. The extracted DNA was quality-checked with 260/280 and 260/230 nm wavelength ratios using a NanoDrop™ (Thermo Scientific, Massachusetts, USA) and stored at −20 °C until sequencing. For bacterial amplicons, the universal prokaryote primers 515 F and 926 R were used to amplify the 16S V4 subregion of rRNA gene[77]. DNA samples were amplified using the following conditions in three replicate runs: 95 °C for 15 min, followed by 26 cycles of 95 °C for 30 s, 50 °C for 30 s and 72 °C for 1 min with a final extension step at 72 °C for 10 min. The 25 μl PCR mix consisted of 18 μl sterilized H$_2$O, 5 μl 5 × HOT FIREPol Blend MasterMix 0.5 μl of each primer (20 μl) and 1 μl template DNA (final concentration of 400 nM). For amplicons of eukayotes/fungal ITS, we used the universal eukaryote PCR primers ITS9mun and ITS4ngsuni[78]. PCR amplification followed the protocol described in ref. [79]. Briefly, 0.5 μl of each forward and reverse primer (20 mM), 1 μl of DNA extract and 18 μl ddH$_2$O were used in combination with 5 μl of 5 × HOT FIREPol Blend Master Mix (Solis Biodyne, Tartu, Estonia). Thermal cycling followed an initial denaturation at 95 °C for 15 min; 25–30 cycles of denaturation for 30 s at 95 °C, annealing for 30 s at 57 °C, elongation for 1 min at 72 °C; final elongation at 72 °C for 10 min; and storage at 4 °C. Duplicate PCR products were pooled and quality checked on a 1 % agarose gel. All primers were tagged with a 10-base pair barcode for sample identification. Blanks containing ddH$_2$O instead of DNA template were used as negative controls in the library preparation. The amplicons from the replicates were pooled, purified using a purification kit containing agarose gel (FavourPrep Gel/PCR Purification mini Kit-300 Preps; Favourgen) and shipped for library preparation in the sequencing service facility of University of Tartu (the Estonian Biocenter). Sequencing of ITS was performed on the PacBio Sequel System at Novogene (UK), and 16S libraries were sequenced on two runs using an Illumina MiSeq platform (2 × 250 bp paired-end chemistry).

Shotgun metagenomic sequencing was performed on pooled equimolar amounts of DNA from a subset (n = 94, or n = 47 pairs) of grassland and forest sites (Table S1). Library preparation and sequencing were performed at the service provider facilities (Novogene Europe) using the Novogene NGS DNA Library Prep Set kit and sequenced on Illumina NovaSeq with 2 × 150 bp paired end reads.

## Bioinformatics

**Metabarcoding.** We used the LotuS2 version 2.22[80] pipeline to quality-filter, demultiplex, and process the filtered reads into OTUs. Chimera detection and removal was done using Uchime[81] with all singletons and sequences shorter than 100 bp discarded. For fungal sequences, the

ITS region was extracted post-clustering, and used to extract and restrict *Blast* searches using ITSx[82]. Clustering of sequences was done using a *de-novo* clustering algorithm in UPARSE[83] based on a 97% similarity threshold. Taxonomy was assigned against the SILVA (ver. 138.1) and UNITE (ver. 8.1) databases for prokaryotic and fungal sequences respectively. All datasets were manually curated to remove contaminant sequences based on negative controls. OTUs representing archaea, chloroplasts, eukaryotes, and mitochondria were omitted from the bacterial dataset, and OTUs unassigned at class level were omitted from the ITS dataset after OTU clustering. To reduce biases due to low sequence coverage, we discarded samples with < 100 reads ($n = 8$ samples) and < 3000 reads ($n = 3$ samples) from the fungal and bacterial dataset, respectively. This resulted in a total of 204 samples comprising 8,532,758 reads (sample mean ± SD = 41,8127 ± 33,514) across 8937 OTUs for bacteria, and 199 samples with a total of 437,006 reads (sample mean ± SD = 2196 ± 2750) covering 2818 OTUs for fungi.

**Shotgun metagenomics.** Analysis of metagenomic reads was done using MATAFILER pipeline[84] using a workflow optimized for complex environmental metagenomes[85]. Briefly, reads obtained from the shotgun metagenomic sequencing of soil samples were quality-filtered by removing reads shorter than 70% of the maximum expected read length (150 bp), with an observed accumulated error > 2 or an estimated accumulated error > 2.5 with a probability of ≥0.01, >1 ambiguous position or if base quality dropped below 20, using sdm (version 1.46)[80]. All 95 samples produced sufficient quantity of reads (average 37 186 296 ± 7 365 270 reads per sample) and were retained for statistical analyses. To estimate the functional composition of each sample, similarity search using DIAMOND (version 2.0.5; options -k 5 -e 1e-4 –sensitive) in blastx mode[86] was employed. Prior to that, the quality-filtered read pairs were merged using FLASH (version 1.2.10)[87]. The mapping scores of two unmerged query reads that mapped to the same target were combined to avoid double counting. In these cases, the hit scores were combined by averaging the percent identity of both hits. The best hit for a given query was based on the highest bit score and highest percent identity to the subject sequence. Using this method, we calculated the relative abundance of (clusters of) orthologous gene groups (OG) by mapping quality-filtered reads against the eggnog database (version 5)[88]. We used the MicrobiomeCensus pipeline to estimate average genome size (AGS) of bacterial communities[89].

**Inferring pathways related to C-N-P cycling.** Quality-filtered reads were blasted against the CAZyme database[90] to derive OGs related to C-cycling. These were further mapped against CAZymes with known substrate affinities based on a custom database provided in ref. 91. For N- and P-cycling genes, we annotated genes from the Kyoto Encyclopedia of Genes and Genomes (KEGG) Orthology database[92] and mapped these against custom databases containing N- and P-cycling genes provided in refs. 93,94. The total number of metagenomic reads, including reads and OG for each nutrient cycling process are shown in Table S4.

**Predicted metagenomes.** Testing for trade-offs between functional redundancy and specialization require joint taxa-function matrices to assess functional distances between taxa[39]. We used *Picrust2* (ver. 2.5.2)[40] to predict functional metagenomes (PFM) from quality filtered 16S rRNA sequences. The weighted nearest sequenced taxon index (NSTI) of the PFM ranged between 0.129 and 0.312, indicating generally good predictions. From the PFMs, we next extracted the functional pathways related to C-N-P cycling based on OG from the KEGG database for subsequent analyses.

## Statistical analyses
All statistical analyses were performed using R (ver. 4.4.1) using the following packages *effectsize*[95], *vegan*[96], *adiv*[97], *lme4*[98], *microniche*[99],

*rfpermute*[100], *SRS*[101], *glmm.hp*[102]. For all analyses involving multiple comparisons, *p*-values were adjusted with Benjamin-Hochberg corrections.

## Diversity analyses
We used Shannon's *H'* to estimate taxonomic (i.e., OTU) and functional (C-N-P-cycling genes) alpha diversity as it considers both richness and evenness. We used non-parametric tests and generalized linear mixed models (LMM) to test for differences in diversity between land-use stages. Specifically, we used Kruskal-Wallis tests to assess differences in diversity between grassland sites, and LMMs to test for differences between paired grassland and forest sites with sample pair and spatial distance between paired sites included as random factors. As forests are the putative successional endpoint of the abandoned grasslands, we set these as the reference level in the LMMs, with subsequent pairwise comparisons in a separate model without a set reference level to examine differences between grassland stages. Two-way ANOVA tests were used to assess whether taxonomic and functional diversity differed depending on differences in forest stand age and evenness, including their interaction. Differences in community composition (beta diversity) were tested using perMANOVA ($10^4$ permutations) on Bray-Curtis distances for taxonomic and functional matrices. To estimate the magnitude of the diversity effects (alpha, beta) between grassland and forest sites, we derived effect sizes (partial $\omega^2$) between paired grassland and forest sites for each grassland category. Effects of forest stand age and evenness on microbial taxonomic and functional diversity were assessed using two-way ANOVA models, including an interaction term of the two factors. Corresponding analyses for community and functional gene composition were done using perMANOVA analyses ($10^4$ permutations) based on Bray-Curtis distances. We used random forest models with combined permutation tests to extract the main covariates influencing microbial taxonomic and functional diversity, including community and functional composition. Based on this variable selection process, we next used hierarchical partitioning of fixed effects in the LMMs to partition the effects of the covariates on diversity variables. Prior to all diversity analyses, OTU matrices were normalized using scaling with ranked subsampling (SRS)[101], with a minimum sequencing depth (*cmin* = 200 and 3000) for fungi and bacteria, respectively. Functional matrices were normalized by rarefying to minimum sequencing depth.

## Assessing functional redundancy and specialization, and their links to carbon-cycling capacity
We used a set of approaches to examine the genetic redundancy related to nutrient cycling during succession. Firstly, taxonomic diversity (Shannon's *H'* of OTU tables) was used as a predictor of functional diversity (Shannon's *H'* of C-N-P-genes from metagenomes) in ordinary least-square regression (OLS) models. For these analyses, all grassland stages were pooled into a single grassland category to match the sample size of forest sites.

Significant fits between taxonomic and functional diversity was deemed indicative of low functional redundancy[49]. This reasoning follows the assumption that functional redundancy ensues from the accumulation of taxa sharing the same effect (i.e., metabolic) trait[103]. Removing taxa from the community will not immediately decrease the diversity of functional genes in systems with high redundancy. In systems with low redundancy, the opposite is true, and the two variables are consequently significantly related. We next used Levin's niche overlap[104] (implemented in the *Microniche* package[99]) to quantify the metabolic overlap of C-N-P genes across land uses, using rarefied metagenome matrices. Only genes above the default limit of quantification (LOQ = 1.65) were kept for subsequent analyses. We assumed that genes with high overlaps are occur widely within a given land use, akin to generalist taxa in classical niche theory, whereas low overlaps indicate specialization[104]. The analyses were repeated using specific

nutrient cycling pathways. We used OLS and second-order polynomial regressions to assess changes in average genetic overlap across successional stages, with pairwise Wilcoxon tests were used to test for differences in average overlap of substrates (C-cycling) and nutrient cycling pathways (N-P-cycling) between land use-stages. Differences in SIR between paired grassland and forest sites were tested through effect size measurements (Hedge's *d*) for each assessed substrate. We then examined the relationship between the communities' total carbon cycling capacity and diversity by regressing MSIR as response variable against taxonomic and functional diversity as predictor variables in OLS models.

### Assessing trade-offs between functional redundancy and diversity

Trade-offs between community-level functional redundancy and diversity, as well as drivers of both measures, were examined with predicted bacterial metagenomes based on the framework developed in refs. 38,39. We considered KO pathways related to C-N-P-cycling as traits and related these to the relative abundances of bacterial OTUs at each site. Specifically, we first scaled all pathways (range 0-1) by their minimum and maximum values and calculated the functional Euclidean distances between all pairs of OTUs. This step was done for each land-use stage separately to avoid the inclusion of OTUs not found within the meta-community. Functional distances were then scaled by division with their land-use specific maximum values. From this, functional diversity (Rao's quadratic diversity $Q$), functional redundancy $R$, and the Simpson dominance index $D$ were calculated using the *adiv* package[97]. The position of each site along the axes of these three measures were assessed and visualized using ternary plots and perMANOVA tests (Bray-Curtis distances, 9999 permutations). Note that the limited availability of fungal genomes precludes similar analyses for the fungal dataset.

### Reporting summary

Further information on research design is available in the Nature Portfolio Reporting Summary linked to this article.

## Data availability

The sequence data generated in this study have been deposited in the NCBI database (amplicon sequences) under accession PRJNA1238768, and in SRA (shotgun sequences) under the project ID PRJEB56463. The 16S and ITS metabarcoding data generated in this study have been deposited in the NCBI Sequence Read Archive (SRA) under the Bio Project accession number PRJNA994701. The carbohydrate active enzymes (CAZy) Database is available under (http://www.cazy.org/). The nitrogen cycling gene (NCyc) Database is available under (https://github.com/qichao1984/NCyc) (https://doi.org/10.1093/bioinformatics/bty741), the phosphorus cycling gene (PCyCDB) Database is available under (https://github.com/ZengJiaxiong/Phosphorus-cycling-database) (https://doi.org/10.1186/s40168-022-01292-1). The Evolutionary genealogy of genes: Non-supervised Orthologous Groups (EggNOG) Database is available under (http://eggnog5.embl.de/#/app/home). The Kyoto Encyclopedia of Genes and Genomes (KEGG) Database is available under (https://www.genome.jp/kegg/kegg1.html). Picrust2 (ver. 2.5.2) was used to predict bacterial metagenomes and is available under (https://huttenhower.sph.harvard.edu/picrust/). Data and code to reproduce the figures are available at: (https://zenodo.org/records/17176048).

## Code availability

Data and code to reproduce all analyses and figures after bioinformatic processing are accessible on Github under the following repo: (https://github.com/tranheim/Succession_functional_diversity/tree/main) and (https://zenodo.org/records/17176048).

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

## Acknowledgements

We wish to thank Anders Glimskär for his contribution with finding sites and providing metadata. Part of computations and data handling were enabled by resources provided by the National Academic Infrastructure for Supercomputing in Sweden (NAISS), partially funded by the Swedish Research Council through grant agreement no. 2022-06725. The following authors acknowledge funding enabling this research: TRS: C.F. Lundströms Stiftelse, CF2023-0019, Lars Hiertas Minne FO2021-0302; MB: Swedish University of Agricultural Sciences (early career grant), the Swedish Research Councils Formas (Grant 2020–00807), the Swedish Research Council (VR; Grant 2021–03724) and Novo Nordisk Foundation (NNF24OC0089849); FH & JF: were supported by the UKRI Biotechnology and Biological Sciences Research Council (BBSRC) Institute Strategic Programme Food Microbiome and Health BB/X011054/1 and its constituent project BBS/E/F/000PR13631, Decoding Biodiversity BBX011089/1 and its constituent work packages BBS/E/ER/230002 A and BBS/E/ER/230002B, FH as well as by European Research Council H2020 StG (erc-stg-948219, EPYC); JF: the BBSRC Norwich Research Park Biosciences Doctoral Training Partnership, BB/T008717/1.

## Author contributions

T.R.S., M.V., J.B., S.E.H., J.S., and M.B. conceptualized the study. T.R.S., J.F., F.B., K.P., J.L., E.O., and F.H. collected the data. T.R.S. and M.B.

analyzed the data. T.R.S., J.B., M.V., and M.B. wrote the manuscript. T.R.S., M.V., J.B., K.P., E.O., and M.B. reviewed and edited the manuscript. T.R.S. and M.B. revised the manuscript. M.B. supervised the study.

## Funding

## Competing interests
The authors declare no competing interests.
