## [Transparent Peer Review file · Nature Communications]

Functional diversity of soil microbial communities increases with ecosystem development

Corresponding Author: Dr Tord Ranheim Sveen

Version 0:

Reviewer comments:

Reviewer #1

(Remarks to the Author)

This study found that in the process of grassland to forest succession, soil microorganisms shifted from high taxonomic diversity with functional redundancy to lower taxonomic diversity with functional specialization. Succession enhanced the specialization of microbial C-N-P cycling genes while decreasing genetic redundancy. This finding provides insights into the functional changes in soil microbial communities following land abandonment. The study is logically rigorous, well-supported by evidence, and offers valuable conclusions. However, certain details require clarification and improvement, as outlined below:

In Fig. 1a, the study site is described as "a national successional gradient in Sweden", but the forests in this study are actually "conventionally managed forest sites". Is it appropriate to consider conventionally managed forest sites as the final stage of succession in this study? What is the justification for this choice?

The abbreviation in appendix figures lack detailed explanations. Please provide additional clarifications. In addition, "C:N" is incorrectly written as "C.N" in the appendix.

No data on fungal genes related to N-cycling is provided. Could this impact the study's conclusions?

In L198-L201, why is AGS included as a fixed effect in all bacterial models? Table S8 does not appear to support the claim that AGS "did not alter the direction or significance of the taxonomic-functional diversity relationships". Based on L317, this should likely refer to Table S10 instead.

The notation "Fig. **" is inconsistently written as "Figs. **" in several places (e.g., L187, L190, L220). Please ensure consistency throughout the paper.

The reference to Fig. S5 in L249-L250 appears incorrect; it likely refers to Fig. S6. There are also some figure and table references do not align with the content. For instance, Table S6 in L123 should be Table S7. Please verify all references carefully.

The discussion in L319-L323 lacks sufficient supporting evidence and appears unconvincing. If this argument is to be maintained, additional supporting literature should be cited.

In L348-L349, multiple important factors were identified in Fig. S6. Why are only soil C:N, pH, and LDMC discussed? These three factors were also highlighted in L292-L293. Total C increased significantly in forests (Table S3) and was identified as important in Fig. S6. Why was it not included in the discussion?

The conclusion does not comprehensively summarize the key findings and should be refined.

In L423-L424, the incubate time is only 4 hours. What is the basis for this?

In L558-L559, only bacterial data were statistically analyzed to examine the trade-off between functional redundancy and diversity, while no similar analysis was conducted for fungi. Therefore, the discussion in the paper should explicitly state that the results apply to bacteria rather than broadly referring to "microorganisms".

(Remarks on code availability)

Reviewer #2

(Remarks to the Author)

Summary:

This study conducted an impressive and large scale sampling of geographically clustered sites with different land uses arranged from managed grasslands, recently abandoned grasslands, late successional grasslands, to afforested forests. They performed DNA metabarcoding, metagenomics, edaphic characterisations, and substrate induced respiration, and included a very nice and detailed set of statistical analyses to explore variation in biodiversity, potential microbial niche overlap, and functional gene variations. They show some major deviation between land use types with respect to microbial richness, community composition, functional gene relative abundances, and what is most exciting to see, correlations between microbial biodiversity metrics and the relative abundance of functional genes relevant for biogeochemical cycles. These latter correlations are especially interesting because they help to broadly define the links between microbial biodiversity and potential ecosystem function. While many of the results are exciting and well presented in effective display items, there are some important factors to consider in how the authors portray and report the results. They report strong results with causal language. And while this is bold and thought-provoking, it is also a significant decision to interpret their observational results with causal interpretation. I think this requires significant revision and reconsideration. There are also fundamental theoretical issues in how the study is framed around succession towards an afforested end point which I discuss in more detail below. Finally, there are also important methodological decisions with soil sampling and statistical analyses that further challenge interpretation of the results through the narrow lens of "ecosystem development" which the authors push. I also discuss these in my high level feedback below. For these reasons, I think the manuscript requires significant reconsideration before publication even though I see a nice potential for this work.

High level feedback:

1. I am not convinced the authors can say that communities and their functions are changing with succession with the degree of confidence as presented in its current form in the manuscript. This is because the sampling design is comparative and based on observation with a few key factors differentiating the samples collected from different land use types. The most obvious extraneous factor being the authors compare samples from only the mineral horizon (grasslands) to samples with an organic horizon and possibly also a mineral horizon (forest sites). Since coniferous forests in Sweden often have deep organic horizons, this could produce a comparing "apples to oranges" scenario. This limitation is also not addressed anywhere in the manuscript.
2. The authors often report non-linear changes across the land uses organised from managed to afforested (which they frame as succession). While this suggests that the results may be valid reflections of different land use types, I am not convinced the patterns necessarily reflect a pattern driven by continuous successional dynamics towards the afforested sites.
 - 2a. There is a lack of continuity in the directionality for many responses. And while the authors attempt to explain this via threshold responses (an interesting and valid interpretation), there may be other explanations related to other variables that did not receive attention in the manuscript. The directional pattern of some changes across their hypothetical successional gradient are also not consistent (e.g., fungal C cycling genes, bacterial P cycling genes, bacterial N cycling genes), which further questions whether this is really thresholds or variation across land use types with the most distinct land use type (afforested) often being most dissimilar to the grasslands.
 - 2b. Many of the results further show extreme heterogeneity across space, especially results from the analyses of niche overlap, Shannon diversity, and patterns of community composition. This is very interesting variation that is likely captured by other co-variables, like you show in your RF analyses. Perhaps analysing each of the non-forest characteristics relative to the "paired" afforested site and seeing if this effect size deviates from managed, recent, and late in an expected manner could help. This nevertheless has two major implications.
 - 2c. Is the land use effect size large compared to other effects of other co-variables and is it independent?
 - 2d. Unless land use management would convert grasslands into forests, there is also a logical gap in the general framework of the study that grasslands would develop continuously into forests without explicit management. They are afforested sites, which means the land use was not as a forest for a very long time (or possibly never), and the later successional grasslands would not become forests unless managed to become forests, which is not really development of an ecosystem based on succession but rather managed succession. Maybe revise wording or framing accordingly.
3. I encourage the authors to be significantly more cautious with causal language throughout the text, as they consistently use terms like X drove Y and X affected Y. I think your observational approach is very important and holds great promise towards understanding variation in land use types across Sweden, but it requires appropriate language to describe patterns and correlations which you currently do not do well in the manuscript. These types of causal relationships cannot be teased apart in your study, and to my knowledge, many co-variables that can affect microbial communities were also not included in your main models to account for their effects (e.g., the LMMS did not include co-variables like soil pH, C:N ratio, etc...). The manuscript contains this language throughout, including in section titles, and it should be toned back to accurately reflect your study design and results. Are your results still consistent if you include co-variables you used in your RF models in the LMMS?
4. The discussion in its current form is short and snappy, and that is quite nice, but it often and unfortunately has a fair amount of shallow interpretation and is filled with causal language, including expressions of direct causality (e.g., L339). There is a lot of jargon and absence of details when referring to other published literature, and it would enrich the discussion if you elaborated versus only briefly touched upon results from other studies. Examples of this shallow interpretation are shown at L303, 306, L331, L334, L345, L355. They are single sentence descriptions of results without much/any detail or interpretation. Can you elaborate at some of these sections so your results are put into a broader scientific context?
5. There are also no acknowledged limitations in your study in the main text, but there are many of fundamental importance (see comments above). The correlational nature of the results and the use of PICRUST2 are two important factors important to address in the discussion that I did not see discussed above.

Line-by-line feedback:

1. Introduction:

L68: Specifying 'agricultural' land abandonment would help to provide context for those less familiar with the topic.

L73: Can we be sure that abandoned land, when left to its own devices, will regenerate into a full forest ecosystem? And is it known how long this will take? There needs to be more explanation of the aboveground successional trends.

L82-85: Are you really able to call this a mechanistic link or speak with such confidence around causality? I ask because these ideas are supported by correlational patterns in an observational study, but you state these trends as causal. You also begin this paragraph stating mechanisms. I would encourage you to tone back a bit here.

L95: An alternative interpretation would be that these processes are not always well captured by functional gene relative abundances.

L96: High functional redundancy with respect to specific traits but not necessarily the actual functional capacity of species if we consider their wider functional capabilities and niches.

L99: Afforestation is a broad term. Are you looking at succession through natural afforestation or planted afforestation? There is a disconnect here because you use a reference forest that was presumably planted, but then your grasslands are naturally regenerating.

Additionally, it is not explained what processes cease following land abandonment (i.e., compaction, grazing, fertilization), so it is difficult to determine why microbial communities might change between the first two stages.

L103: I would not call this "biogeochemical potential" but rather "functional genes encoding enzymes involved in soil biogeochemical cycles". Same at L106, I would not say "biogeochemical redundancy" as this is not accurate.

L105: What do you mean by specialisation of microbes? Since this is a hypothesis, I think the predictors and response variables should be more explicitly defined.

2. Methods:

L382: Can you justify how the sites can be paired based solely on geographic distance? Did you look to see if this is justified based on other factors like soil classification, elevation, depth of the water table, nitrogen deposition, or something else? I am not asking for you to address each of these points, but rather to better elaborate on the pairing decision since geographic distance may not be able to accurately classify sites as paired or perhaps, it is sufficient. Details can help us better understand and evaluate how much we buy into the design.

L391: Can you explain why you make this assumption?

L400: So at the forest cover sites, you are comparing organic horizons to mineral horizons samples at the grassland and agricultural sites?

L414: Can you explain why only 120 of the 235 plant species were used?

L427:

a. Typo around here?

b. Substrate induced respiration is used to measure microbial biomass, not carbon cycling capacity (Anderson and Domsch, 1978). Can you justify this framing? I suggest you revise because some of your key interpretations may be based on variation in biomass more than C cycling potential.

L465: Did you extract the fungal ITS region prior to clustering? There is a variable lengthened portion of the SSU and LSU in the amplicons when using these primers, and it can influence your taxonomic assignments and sequence similarity threshold decision making.

L469: How do you justify removing OTUs not assigned beyond the class level?

L476: It's great to see you do direct read mapping! How successful were you with assigning reads to fungi in the metagenomes? Can you provide some details about how many sequences were assigned to fungi versus prokaryotes?

524: How were the effect sizes measured? This will really help with interpreting Figure 2.

L535: This is an interesting idea. Could you add 1-2 more sentences explaining why you interpret this correlation this way?

L549: Why is this a tradeoff?

3. Results:

L118: But diversity was lower so would it not be the inverse of Figure 1e?

L126 & L127: Missing references to Fig. 2e.

Figure 2: What is the horizontal line in panel e?

L132-134: Great work including this!

L189-192: Very interesting result!

L213: Can you really say: "a gradual specialization across the entire soil microbial communities during ecosystem development". To me, this is a discussion of a result. Rather, as a result, you can say across the land uses, right?

L225: From my understanding, this section only analysed bacteria, so that needs to be specified in the title rather than referring to 'microbes' which could be taken as fungi and bacteria.

Similarly, the figure caption in Figure 5 also needs to specify that this is only about bacteria. Including a bacteria symbol in the figure would also help this interpretation.

L235: Is this a distinct hypothesis from what you propose in the introduction?

L244: Figure S6b?

Fig. 4: What do you mean by increasing molecular complexity? What is the vertical axis in panels e-f?

Fig 5. The figure caption does not refer to panel c.

L260: I think you need to be clearer about what you measured with the SIR and how you present your results (see my comment above). I would discourage referring to this as "C-cycling rates" as that is not only a very general description that could mean almost anything, but it's also probably inaccurate when describing SIR (see comment in methods). I would rather say, "substrate induced respiration of...", similar to how to refer to it in figure 6. You go on to even call this decomposition at L269, so there is a very high degree of looseness applied to your interpretation of the SIR results.

L262: Were these differences significant? I admit I do not see this result

L267: This is an interesting results and nice correlation to see.

4. Discussion:

L283-284: You cannot examine how microbes respond to land abandonment in this study. You are using a space-for-time substitution, and there are other factors that co-vary with your design which may not have to do with succession, such as your sampling and comparison of mineral soils in the grasslands to soils containing the organic horizon in forests. I would encourage a more accurate and honest description of what your study is able to address and to avoid over-selling.

L284-290: I do not see the threshold responses as clearly as you seem to propose across Figs 2-4. In Fig. 2, this patterns seems only present in panel a; in Fig. 3, this patterns only seems visible in panel b; and in Fig. 4 this patterns only seems visible in panel c. Thus, across most scenarios, this patterns is not present.

L298: Why would these enzymes be “complex”?

L305-306: I am unsure if this sentence is necessary or accurate. Climatic events are very different processes to land abandonment, and Knight et al. 2024 does not incorporate succession. There are also no responses in your study so I am not sure this reference supports your point. Also what is the point you are trying to make here?

L310: You mean correlation versus relationship?

L325: This refers to alpha and beta diversity, but many of your results are more complicated calculations, so I would be more specific about what you are talking about. Here, you should say what you really show in the figure: that climate was not correlated with microbial alpha and beta diversity.

L342: You say that functional diversity and functional redundancy may be inversely correlated, but does this contradicts your earlier results (Fig. 3) where bacterial diversity was positively correlated with functional genes, and fungi showed only a negative correlation within forests?

L349: I am still not sure how you measured resource complexity in this study, and can therefore call this a driver of your results.

L358-366: Can you add these details earlier since they provide key justifications.

5. References

L581: Reference 6. This reference is incomplete and Arias-Navarro's name is misspelt.

L756: Reference 84. Title needs spaces fixing.

(Remarks on code availability)

Reviewer #3

(Remarks to the Author)

(Remarks on code availability)

Version 1:

Reviewer comments:

Reviewer #1

(Remarks to the Author)

This study found that in the process of grassland to forest succession, soil microorganisms shifted from high taxonomic diversity with functional redundancy to lower taxonomic diversity with functional specialization. Forest succession enhanced the specialization of microbial C-N-P cycling genes while decreasing genetic redundancy. This finding provides insights into the functional changes in soil microbial communities following land abandonment. Overall, the study is logically rigorous and well-supported by the evidence. Moreover, all the comments from the reviewers and me were fully considered, so I have no more comments in this round. I think it is a intersting and excellent manuscript. I recommend it to be published on Nature Communications.

(Remarks on code availability)

Reviewer #2

(Remarks to the Author)

This is a very well revised manuscript, and I applaud the co-authors for so carefully addressing each point raised during the initial review. The resulting manuscript has more depth, nuance, and it still captures its original and important point while placing the results into a richer context. The discussion is really much better and very interesting.

Incorporating additional analyses to further explore your points (and including additional data on stand age and evenness) demonstrate a great effort and dedication. Well done!

I appreciate the additional discussion of the soil sampling x depth issue I raised in the methods section. While it remains an outlying “problem”, and I am not convinced it doesn't affect interpretation of the results, I also understand that the authors feel

this best aligns with their study goals. I admit, I am not so convinced because comparing soil with and without organic horizons and with varying levels of organic horizon material will shape every soil analysis. I nevertheless do not think nor suggest that this should prevent publications. If it were my paper, I would feel it necessary to add a statement in the discussion squarely saying something around the lines of “by sampling at a standard soil depth, we compare samples with and without organic horizons and with differing organic horizon depths, which strongly shapes resource availability, mineralogy, and physicochemical conditions that would be better captured by sampling at a consistent 0-10 cm depth in the mineral horizon”. This would go well somewhere around L952. I leave it to the editor to decide whether this important or not.

I also appreciate that the authors toned down causality (in many places), and discuss their results extremely effectively (great job with the revised work at L743-815). Nevertheless, there is still a considerable amount of causal language in the revised text. I would personally feel more comfortable, if this were my paper, to try and reduce this even further (e.g., at L44, 264, 265, 484, 486, etc..). Again, I leave this to the editor to design if it warrants additional attention.

Good job on the revision. I look forward to seeing your work published.

(Remarks on code availability)

Reviewer #3

(Remarks to the Author)

(Remarks on code availability)

Reviewer comments

Reviewer #1 (Remarks to the Author):

This study found that in the process of grassland to forest succession, soil microorganisms shifted from high taxonomic diversity with functional redundancy to lower taxonomic diversity with functional specialization. Succession enhanced the specialization of microbial C-N-P cycling genes while decreasing genetic redundancy. This finding provides insights into the functional changes in soil microbial communities following land abandonment. The study is logically rigorous, well-supported by evidence, and offers valuable conclusions. However, certain details require clarification and improvement, as outlined below:

Answer: We thank the reviewer for the positive assessment of our manuscript and the constructive feedback. We have addressed their comments by clarifying our methodology and improving the discussion.

In Fig. 1a, the study site is described as “a national successional gradient in Sweden”, but the forests in this study are actually “conventionally managed forest sites”. Is it appropriate to consider conventionally managed forest sites as the final stage of succession in this study? What is the justification for this choice?

Answer: This is an important point which coincides with Reviewer #2’s concerns below. We have provided additional, extensive explanations justifying the choice of conventionally managed forests as the reference point for abandoned grasslands, and additionally included new analyses showing that forest age and evenness, good proxies for forest management, do not affect the results obtained in the study. These are discussed on lines 86-90, and l.724-726. Additionally, our supplementary analyses using metadata on forest stand age and evenness show that management does not appear to be a main factor influencing our results (Table S8-S9; l.775-787).

The abbreviation in appendix figures lack detailed explanations. Please provide additional clarifications. In addition, “C:N” is incorrectly written as “C.N” in the appendix.

Answer: We have added additional explanations for all figures in the supplementary materials and fixed the C:N issue.

No data on fungal genes related to N-cycling is provided. Could this impact the study’s conclusions?

Answer: This is an interesting question to which our answer is unfortunately limited by the lack of databases containing fungal N-cycling genes. While fungi do perform nitrification, the enzymes and pathways involved are poorly understood, and most identified fungal N-cycling genes relate to nitrate/nitrite reductase (i.e. the nirK gene) in denitrification, which is not likely to be a key process in these natural and unfertilized systems. We have revised the manuscript (l. 912-914) to clarify that the database on fungal N-cycling genes does not exist, and that mapping success was poor.

L198-L201, why is AGS included as a fixed effect in all bacterial models? Table S8 does not appear to support the claim that AGS “did not alter the direction or significance of the taxonomic-functional diversity relationships”. Based on L317, this should likely refer to Table S10 instead.

Answer: This is correct; lines 198-201 should indeed refer to Table S10. AGS was included as a fixed effect to account for the fact that in bacteria, the number of genes scales linearly with the size of the genome¹, meaning that the increase in genetic diversity could be simply due to

bacteria in forests having larger genomes. This is not the case here. We have changed the text accordingly.

The notation “Fig. *” is inconsistently written as “Figs. *” in several places (e.g., L187, L190, L220). Please ensure consistency throughout the paper.

Answer: These have now been corrected.

The reference to Fig. S5 in L249-L250 appears incorrect; it likely refers to Fig. S6. There are also some figure and table references do not align with the content. For instance, Table S6 in L123 should be Table S7. Please verify all references carefully.

Answer: We appreciate the careful reading and noticing of these errors. We have revised the manuscript carefully to ensure consistency between the text and all figures and tables.

The discussion in L319-L323 lacks sufficient supporting evidence and appears unconvincing. If this argument is to be maintained, additional supporting literature should be cited.

Answer: As noted above, the lack of reliable makes it challenging to explicitly incorporate fungal genome size in the analyses. However, we have now expanded our discussion to provide a better line of arguments supporting the idea that increasing fungal genome size with afforestation may underpin the observed negative correlation in Fig. 3i.

In L348-L349, multiple important factors were identified in Fig. S6. Why are only soil C:N, pH, and LDMC discussed? These three factors were also highlighted in L292-L293. Total C increased significantly in forests (Table S3) and was identified as important in Fig. S6. Why was it not included in the discussion?

Answer: It is true that total C also play an important part according to the variable selection analyses, but also in the newly added hierarchical partitioning analyses (see e.g. Fig. S2b). This definitely merits a part in the discussion, which we have now added on lines 766-767.

The conclusion does not comprehensively summarize the key findings and should be refined.

Answer: We have now rewritten the conclusion to better summarize the findings of the study (lines 1020-1027).

In L423-L424, the incubate time is only 4 hours. What is the basis for this?

Answer: Incubation time varied depending on substrate applied, with glucose, glycine, oxalic acid, and autolyzed yeast incubated for 4 hrs and lignin and chitin incubated for 24 hrs. For basal respiration, incubation time was similarly 24 hrs, which is quite standard². The method is essentially a multiple substrate-induced respiration method and follows a protocol established at the Netherlands Institute of Ecology (NIOO)³, which has been used to measure soil respiration and biomass in e.g. isotope-based studies⁴. As such, it is comparable to the EcoPlate and MicroResp methods⁵, which are widely used for microbial ecophysiological profiling using multiple carbon substrates. We have now clarified this in the methods on line 1131.

In L558-L559, only bacterial data were statistically analyzed to examine the trade-off between functional redundancy and diversity, while no similar analysis was conducted for fungi.

Therefore, the discussion in the paper should explicitly state that the results apply to bacteria rather than broadly referring to "microorganisms".

Answer: We fully agree with this and have re-written the relevant discussion part accordingly.

Reviewer #2:

This study conducted an impressive and large scale sampling of geographically clustered sites with different land uses arranged from managed grasslands, recently abandoned grasslands, late successional grasslands, to afforested forests. They performed DNA metabarcoding, metagenomics, edaphic characterisations, and substrate induced respiration, and included a very nice and detailed set of statistical analyses to explore variation in biodiversity, potential microbial niche overlap, and functional gene variations. They show some major deviation between land use types with respect to microbial richness, community composition, functional gene relative abundances, and what is most exciting to see, correlations between microbial biodiversity metrics and the relative abundance of functional genes relevant for biogeochemical cycles. These latter correlations are especially interesting because they help to broadly define the links between microbial biodiversity and potential ecosystem function. While many of the results are exciting and well presented in effective display items, there are some important factors to consider in how the authors portray and report the results. They report strong results with causal language. And while this is bold and thought-provoking, it is also a significant decision to interpret their observational results with causal interpretation. I think this requires significant revision and reconsideration. There are also fundamental theoretical issues in how the study is framed around succession towards an afforested end point which I discuss in more detail below.

There are also fundamental theoretical issues in how the study is framed around succession towards an afforested end point which I discuss in more detail below. Finally, there are also important methodological decisions with soil sampling and statistical analyses that further challenge interpretation of the results through the narrow lens of “ecosystem development” which the authors push. I also discuss these in my high level feedback below. For these reasons, I think the manuscript requires significant reconsideration before publication even though I see a nice potential for this work.

Answer: We thank the reviewers #2 & #3 for the positive assessment and the depth of engagement with our study, as well as for their constructive feedback. We address their comments point-by-point below.

1. I am not convinced the authors can say that communities and their functions are changing with succession with the degree of confidence as presented in its current form in the manuscript. This is because the sampling design is comparative and based on observation with a few key factors differentiating the samples collected from different land use types. The most obvious extraneous factor being the authors compare samples from only the mineral horizon (grasslands) to samples with an organic horizon and possibly also a mineral horizon (forest sites). Since coniferous forests in Sweden often have deep organic horizons, this could produce a comparing “apples to oranges” scenario. This limitation is also not addressed anywhere in the manuscript.

Answer: We believe that this comment may partly stem from a misunderstanding where we have not been clear enough in our method descriptions. The soils were sampled in similar ways, that is, 0-10 cm, independent of land use, and therefore typically include both organic and mineral horizons. When needed, loose litter lying on the surface of the sites, typically in forests but also in grasslands, was removed prior to sampling to avoid filling the core with undecomposed litter. While this means that some peaty forest soils containing no mineral horizon were included (a total of 12 sites, or 11 % of all forest soils), a total of 7 grassland soils (6.8 % of all grassland sites) were also peat soils with no mineral horizon in the 0-10 cm range investigated here. These grasslands were moreover distributed across all successional stages. The majority of soils for both land-use types were located on moraine parent material (60 % of all sites investigated) and therefore included both organic and mineral horizons in

our sampling. While patterns may deviate between soil horizons within a single soil and between soil types, we consider that analysing the general patterns across multiple soil types at a standardized depth better corresponds to the overall aim of the study to examine large-scale land abandonment and afforestation which occurs at national and continental scales, spanning a broad range of parent materials and soil types. We have added a paragraph in the method section (lines 1107-1109) to better clarify the sampling procedure. In addition, we have highlighted the potential limitations of our study at several parts in the discussion section (e.g. l.777-787, l.955-957, l.975-981).

2. The authors often report non-linear changes across the land uses organised from managed to afforested (which they frame as succession). While this suggests that the results may be valid reflections of different land use types, I am not convinced the patterns necessarily reflect a pattern driven by continuous successional dynamics towards the afforested sites.

Answer: We thank the reviewers for this comment. Since the points raised by the reviewers under this comment were related, we respond to them in one section (please see below).

2a. There is a lack of continuity in the directionality for many responses. And while the authors attempt to explain this via threshold responses (an interesting and valid interpretation), there may be other explanations related to other variables that did not receive attention in the manuscript. The directional pattern of some changes across their hypothetical successional gradient are also not consistent (e.g., fungal C cycling genes, bacterial P cycling genes, bacterial N cycling genes), which further questions whether this is really thresholds or variation across land use types with the most distinct land use type (afforested) often being most dissimilar to the grasslands.

2b. Many of the results further show extreme heterogeneity across space, especially results from the analyses of niche overlap, Shannon diversity, and patterns of community composition. This is very interesting variation that is likely captured by other co-variables, like you show in your RF analyses. Perhaps analysing each of the non-forest characteristics relative to the “paired” afforested site and seeing if this effect size deviates from managed, recent, and late in an expected manner could help. This nevertheless has two major implications.

2c. Is the land use effect size large compared to other effects of other co-variables and is it independent?

2d. Unless land use management would convert grasslands into forests, there is also a logical gap in the general framework of the study that grasslands would develop continuously into forests without explicit management. They are afforested sites, which means the land use was not as a forest for a very long time (or possibly never), and the later successional grasslands would not become forests unless managed to become forests, which is not really development of an ecosystem based on succession but rather managed succession. Maybe revise wording or framing accordingly.

Answer (2a-d): This is a very interesting set of comments which pinpoint some intrinsic challenges of chronosequences in experimental design generally, and with our design more particularly. We agree that threshold dynamics may be due to differing land-use types rather than continuous successional dynamics, but it is not possible to test this explicitly using chronosequence designs, as these are based on space-for-time substitutions and not repeated measures of the same site over time. That being said, we believe that using realistic

*(managed) rather than theoretical (natural/"climax") endpoints is of preference when assessing the outcomes of land abandonment because the vast majority of abandoned grasslands ultimately end up as managed forests in even-aged stands⁶⁻¹⁰, although we do acknowledge that this likely increases dissimilarity between grasslands and forests and could contribute to the threshold responses observed. To address the reviewers' concern, we derived new metadata from the forest sites about stand age and evenness of the trees, which gives a clear indication of management¹¹, and tested whether our results varied according to these factors. With one exception (fungal taxonomic community composition), they did not (Table S8-S9). We interpret this as a strong indication that forest management does not influence the trajectory of the microbiomes when moving from grasslands to forests, and the thresholds are therefore either "real" thresholds or reflect a gap in the chronosequence between late-stage successional grasslands and fully afforested sites. We have rewritten sections of the result and discussion parts incorporating these new results/insights, while better reflecting the uncertainty regarding thresholds (**point 2a**) and better motivating the choice of managed forest sites as reference endpoints (**point 2d**).*

*As for the more technical parts of the comments (**points 2b-c**), we fully agree with the value of the suggested approaches while noting that a similar approach was already taken for the alpha- and beta diversity analyses shown in Fig. 2a-d and Fig. 3a-e. These were done by comparing the sites within each grassland category to their paired forest reference site, whereas additional tests examined differences between grasslands and were generally not significant (Tables S6, S11). However, we find the suggested approach of calculating effect sizes between paired grassland and forest sites for each grassland category (managed, recent, late) separately promising and have done these analyses with results incorporated into Figs. 2-3 and Figs. S2-S3. In addition, we conducted new analyses using hierarchical partitioning of R^2 values from mixed models to disentangle the relative contribution of covariates (pH, C:N, etc) and succession to the diversity variables examined, while always including pair and distance between pairs as random factors in the models. The result and discussion sections have been rewritten to accommodate these new results. Whether the controlled effect sizes of land use are independent from the effects of covariates (**point 2c**) is a very interesting question, and our answer is: They are likely not independent because land use affects both soil abiotic properties and microbes, which in turn further modify soil properties¹². This is why, in our opinion, effects of land-use change are best analysed at this whole-system level (i.e. as changing ecosystems), whereas underlying drivers can be identified, e.g. through random forest models and the newly added hierarchical partitioning. We have now reflected on these points on lines 767-774.*

3. I encourage the authors to be significantly more cautious with causal language throughout the text, as they consistently use terms like X drove Y and X affected Y. I think your observational approach is very important and holds great promise towards understanding variation in land use types across Sweden, but it requires appropriate language to describe patterns and correlations which you currently do not do well in the manuscript. These types of causal relationships cannot be teased apart in your study, and to my knowledge, many co-variables that can affect microbial communities were also not included in your main models to account for their effects (e.g., the LMMs did not include co-variables like soil pH, C:N ratio, etc...). The manuscript contains this language throughout, including in section titles, and it should be toned back to accurately reflect your study design and results. Are your results still consistent if you include co-variables you used in your RF models in the LMMS?

Answer: We thank the reviewers for this pertinent comment, which we agree with. We have revised the language throughout the manuscript to moderate causal language. As for the co-variables in LMMS, please see our answer to points 2a-d above.

4. The discussion in its current form is short and snappy, and that is quite nice, but it often and unfortunately has a fair amount of shallow interpretation and is filled with causal language, including expressions of direct causality (e.g., L339). There is a lot of jargon and absence of details when referring to other published literature, and it would enrich the discussion if you elaborated versus only briefly touched upon results from other studies. Examples of this shallow interpretation are shown at L303, 306, L331, L334, L345, L355. They are single sentence descriptions of results without much/any detail or interpretation. Can you elaborate at some of these sections so your results are put into a broader scientific context?

Answer: This comment is especially appreciated as we strive to make sense of other results to interpret our own, rather than merely name-dropping. We have made an effort to improve this aspect throughout the entire discussion, especially related to the lines pointed out by the reviewers, see especially the paragraph running from l. 915-961.

5. There are also no acknowledged limitations in your study in the main text, but there are many of fundamental importance (see comments above). The correlational nature of the results and the use of PICRUS2 are two important factors important to address in the discussion that I did not see discussed above.

Answer: This is true, and we have added a paragraph highlighting these limitations in the discussion section (l.547-549, l.978-981).

L68: Specifying ‘agricultural’ land abandonment would help to provide context for those less familiar with the topic.

Answer: Good point. The text has been changed accordingly.

L73: Can we be sure that abandoned land, when left to its own devices, will regenerate into a full forest ecosystem? And is it known how long this will take? There needs to be more explanation of the aboveground successional trends.

Answer: Yes, there is high certainty that abandoned land eventually reaches full afforestation unless other disturbances than management (e.g. fire) reset the successional trajectory, as evidenced by analyses of historical land-use maps⁶⁻¹⁰. The time it takes for an abandoned field/grassland to become fully afforested varies considerably with factors such as climate, soil type, plant metacommunity and colonization and dispersion dynamics, priority effects, etc. and it is therefore difficult to give any precise measures¹³⁻¹⁵. We have added a section in the introduction about successional dynamics during ecosystem development after abandonment (l. 86-90, also l. 174-176).

L82-85: Are you really able to call this a mechanistic link or speak with such confidence around causality? I ask because these ideas are supported by correlational patterns in an observational study, but you state these trends as causal. You also begin this paragraph stating mechanisms. I would encourage you to tone back a bit here.

Answer: This is a good suggestion, which we agree with and appreciate. We have also toned back the purported causality in the discussion part and better highlighted the correlational nature of our results.

L95: An alternative interpretation would be that these processes are not always well captured by functional gene relative abundances.

Answer: This is true. We have now replaced the word “functional” with “genetic” redundancy (l. 109), because this is better in line with the reasoning and the measurements performed. Similar adjustments have been made throughout the manuscript and we have additionally added a sentence better highlighting the potential relation and discrepancies between genetic and functional redundancy in the context of nutrient cycling (l. 993-997).

L96: High functional redundancy with respect to specific traits but not necessarily the actual functional capacity of species if we consider their wider functional capabilities and niches.

Answer: This point aligns well with the comment made above (L. 95), which we address there.

L99: Afforestation is a broad term. Are you looking at succession through natural afforestation or planted afforestation? There is a disconnect here because you use a reference forest that was presumably planted, but then your grasslands are naturally regenerating. Additionally, it is not explained what processes cease following land abandonment (i.e., compaction, grazing, fertilization), so it is difficult to determine why microbial communities might change between the first two stages.

Answer: It is not possible to differentiate between planted and naturally regrown forests without historical source material ranging back to the mid-19th century. The forest sites may additionally have been naturally regrown, cut at some point, and then planted. What we can say with certainty is that our study design reflects “real-world” landscape changes where natural regeneration and plantations are mixed, but the underlying movement is that of large-scale afforestation of abandoned grasslands.

L103: I would not call this “biogeochemical potential” but rather “functional genes encoding enzymes involved in soil biogeochemical cycles”. Same at L106, I would not say “biogeochemical redundancy” as this is not accurate.

Answer: We fully agree with this and have changed the wording accordingly.

L105: What do you mean by specialisation of microbes? Since this is a hypothesis, I think the predictors and response variables should be more explicitly defined.

Answer: We have now rephrased the sentence by referring to the reduced overlap of functional genes instead of specialization.

L382: Can you justify how the sites can be paired based solely on geographic distance? Did you look to see if this is justified based on other factors like soil classification, elevation, depth of the water table, nitrogen deposition, or something else? I am not asking for you to address each of these points, but rather to better elaborate on the pairing decision since geographic distance may not be able to accurately classify sites as paired or perhaps, it is sufficient. Details can help us better understand and evaluate how much we buy into the design.

Answer: We opted for distance-based pairing because the grassland and forest sites derived from environmental monitoring programs and were therefore limited in their distribution. Although by no means perfect, geographic pairing was the best available criterion for pairing as it partly encompasses many of the other covariates such as climate, broad vegetation habitats, N-deposition, geology, etc. whereas pairing by e.g. parent material and water table

depth may conflict with e.g. elevational differences between the sites, leading to other problems. Conversely, satisfying all or multiple criteria simultaneously would have drastically reduced the number of available sites for the study, thereby reducing one of its main aims: to cover land abandonment as it occurs across a national scale. When possible, we have incorporated additional factors (parent material) into the linear models as random factors to explicitly account for this. As stated in the method section (l. 1094-1097), we acknowledge the limitations of distance-based pairing, but we have now rewritten the section about study design in the methods part to clarify this point further.

L391: Can you explain why you make this assumption?

Answer: This relates to the rebuttals made to comments about l. 73 and l. 99 above. We have substantial evidence from historical land-use maps that previously open grasslands, after afforestation, are now managed as production forests. This means that production forests are the best land-use change endpoint for abandoned grasslands. We have highlighted this reasoning in the introduction and discussion sections.

L400: So at the forest cover sites, you are comparing organic horizons to mineral horizons samples at the grassland and agricultural sites?

Answer: See rebuttal to point 1 above.

L414: Can you explain why only 120 of the 235 plant species were used?

Answer: After quality filtering, LDMC data were only available for 120 of the total 235 plant species recorded for grassland and forest sites.

L427:

a. Typo around here?

Answer: We have corrected the sentence (now on l. 426-427).

b. Substrate induced respiration is used to measure microbial biomass, not carbon cycling capacity (Anderson and Domsch, 1978). Can you justify this framing? I suggest you revise because some of your key interpretations may be based on variation in biomass more than C cycling potential.

Answer: We fully agree with the reviewer here that the SIR method was developed for the estimation of microbial biomass. However, by extending the range of substrates used from glucose to other compounds of higher complexity, the method is expanded into a more comprehensive catabolic profiling¹⁶ where the sum of all respiration rates (i.e. the multiple substrate-induced respiration; MSIR) is routinely used to assess total microbial functional capacity^{17,18}. Technically, this is still substrate-induced respiration, although the terminology may lead to confusion as to whether biomass or community-wide capacity to degrade differing substrates is meant. We have revised the wording throughout the manuscript to better clarify this.

L465: Did you extract the fungal ITS region prior to clustering? There is a variable lengthened portion of the SSU and LSU in the amplicons when using these primers, and it can influence your taxonomic assignments and sequence similarity threshold decision making.

Answer: We appreciate the reviewer's concern regarding the potential impact of variable SSU/LSU lengths on our approach. Indeed, we extracted the ITS regions post-clustering. This is the intended purpose of the algorithm of our pipeline, LotuS, for streamlined and efficient analyses of ITS sequences (as the ITS extractor is computationally very intensive, especially for long-read sequences). Importantly, taxonomic assignments were based solely on the ITS

region, as only the extracted ITS sequences were queried against the reference UNITE database. While we acknowledge the theoretical possibility that conserved flanking SSU/LSU regions could influence clustering, this effect is expected to be minimal in practice. The ITS sequence is known to evolve much faster than the highly conserved SSU/LSU rRNA sequence, making it unlikely for divergent SSU/LSU sequences to flank identical ITS regions, or vice versa. Therefore, clustering based on the full amplicon still largely reflects ITS variation. We clarified our approach in the methods on lines 1202-1203.

L469: How do you justify removing OTUs not assigned beyond the class level?

Answer: Since the relationship between taxonomic (i.e. OTU) and functional diversity is of high relevance for the study, we introduced this filtering threshold as a cautionary measure to “nudge” the OTUs toward more defined species boundaries and reduce the risk of falsely inflating taxonomic diversity¹⁹. We have now clarified this in the method.

L476: It’s great to see you do direct read mapping! How successful were you with assigning reads to fungi in the metagenomes? Can you provide some details about how many sequences were assigned to fungi versus prokaryotes?

Answer: Unsurprisingly, there were substantially fewer reads mapped to fungi than prokaryotes in the metagenomes. The read distributions mapped across C-N-P genes for fungi (C-P only) and prokaryotes, together with total accepted reads after quality filtering, are detailed in Table S4 in the supplementary materials. We have clarified this point in the methods.

524: How were the effect sizes measured? This will really help with interpreting Figure 2.

Answer: Effect sizes were derived from LMMs between paired grassland and forest sites (i.e. managed grasslands vs paired forests, recent grasslands vs paired forests, etc) while including pair and geographical distance between pairs as random factors in the models. The figure has now been updated according to the suggested approach (rebuttal to points 2a-d above), and the text and Fig. 2 caption have been amended to reflect this.

L535: This is an interesting idea. Could you add 1-2 more sentences explaining why you interpret this correlation this way?

Answer: Absolutely. We have added more on this assumption, including an improved (more well-known) reference in the method section and when relevant in the discussion.

L549: Why is this a tradeoff?

Answer: Good question. The divergence between functional redundancy and functional diversity (here tightly coupled with specialization) has been previously identified for birds and mammals globally²⁰ and it is in line with this that we referred to it as a trade-off. But given a common definition of the term trade-off (“a situation in which you balance two opposing situations or qualities; Cambridge Dictionary) it is perhaps not perfectly suited to describe the divergence. We have now rewritten this as a “divergent” relationship instead but still refer to trade-off in the discussion (l. 971) to connect the results to the work cited above.

L118: But diversity was lower so would it not be the inverse of Figure 1e?

Answer: Absolutely true. Fig. 1e is only conceptual and may as well show the opposite trend of decreasing patterns. We have added this caveat to the legend of Fig. 1

L126 & L127: Missing references to Fig. 2e.

Answer: This has now been fixed.

Figure 2: What is the horizontal line in panel e?

Answer: It was added to facilitate the visual separation of alpha and beta diversity. See the new modified fig. 2e with the addition of effect sizes as described under the rebuttal to point 2a-d.

L132-134: Great work including this!

Answer: Thank you!

L189-192: Very interesting result!

Answer: We think so, too! In our view, this underpins the idea of a shift from redundancy to specialization.

L213: Can you really say: “a gradual specialization across the entire soil microbial communities during ecosystem development”. To me, this is a discussion of a result. Rather, as a result, you can say across the land uses, right?

Answer: Good point with which we fully agree. We have edited the text accordingly (l. 481-482).

L225: From my understanding, this section only analysed bacteria, so that needs to be specified in the title rather than referring to ‘microbes’ which could be taken as fungi and bacteria. Similarly, the figure caption in Figure 5 also needs to specify that this is only about bacteria. Including a bacteria symbol in the figure would also help this interpretation.

Answer: This is true and we have edited both the text (l. 537) and the figure caption accordingly. We have also included the suggested bacteria symbol in the figure.

L235: Is this a distinct hypothesis from what you propose in the introduction?

Answer: Partly. We realize that the hypotheses were rather poorly stated and have clarified and expanded these now (l. 180-186), including with specific references to them on the appropriate places in the discussion (l. 753-756, l. 939-943, l. 964-965).

L244: Figure S6b?

Answer: Of course; this has been corrected.

Fig. 4: What do you mean by increasing molecular complexity? What is the vertical axis in panels e-f?

Answer: The term “molecular complexity” is perhaps a bit misleading here, as it is really substrate complexity we refer to. We have now changed this in Fig. 4 and Fig. 6. The axis

shows substrates of increasing carbohydrate complexity, requiring more specialized enzymes to degrade.

Fig 5. The figure caption does not refer to panel c.

Answer: Corrected.

L260: I think you need to be clearer about what you measured with the SIR and how you present your results (see my comment above). I would discourage referring to this as “C-cycling rates” as that is not only a very general description that could mean almost anything, but it’s also probably inaccurate when describing SIR (see comment in methods). I would rather say, “substrate induced respiration of...”, similar to how to refer to it in figure 6. You go on to even call this decomposition at L269, so there is a very high degree of looseness applied to your interpretation of the SIR results.

Answer: This is a good point, which we, apart from our reservations, made to the comment on methods, agree with. We now refer to the SIR results as “carbon degradation” or “substrate degradation”.

L262: Were these differences significant? I admit I do not see this result

Answer: The results are significant and are now displayed in Fig. S9.

L267: This is an interesting results and nice correlation to see.

Answer: We think so too, and thus we have now highlighted this result by adding a sentence xxx.

L298: Why would these enzymes be “complex”?

Answer: This is a good point. The enzymes are not necessarily more complex just because the substrates are. The sentence has been modified.

L305-306: I am unsure if this sentence is necessary or accurate. Climatic events are very different processes to land abandonment, and Knight et al. 2024 does not incorporate succession. There are also no responses in your study so I am not sure this reference supports your point. Also what is the point you are trying to make here?

Answer: This was an attempt to generalize our findings more broadly, but it was a rather poor and unsubstantiated attempt. We have now rewritten the entire section and instead elaborated on threshold dynamics with better rooting in relevant literature.

L310: You mean correlation versus relationship?

Answer: Indeed, we mean significant correlation. This has been corrected now.

L325: This refers to alpha and beta diversity, but many of your results are more complicated calculations, so I would be more specific about what you are talking about. Here, you should say what you really show in the figure: that climate was not correlated with microbial alpha and beta diversity.

Answer: This is true. We have rewritten the sentence accordingly (l. 913).

L342: You say that functional diversity and functional redundancy may be inversely correlated, but does this contradict your earlier results (Fig. 3) where bacterial diversity was positively correlated with functional genes, and fungi showed only a negative correlation within forests?

Answer: We thank the reviewers for this comment. As we state earlier on in the discussion (l. 797 in the revised manuscript), it is the absence of a significant correlation itself that indicates functional redundancy, not whether the correlation is positive or negative. If the correlation is (significantly) positive, we will lose functions if we remove taxa. If the correlation is negative, we will gain functions when losing taxa – something that only makes sense if the taxa lost are to some extent replaced by other taxa with higher functional diversity (i.e. specialists), and possibly, larger genomes. While the positive correlations in Fig. 3 are clear enough for bacteria and indicate low functional redundancy, the negative correlation for fungal C-cycling genes is therefore more a sign of increasing specialization – backed up by the decreasing niche overlap in Fig. 4a,e – more than reduced functional redundancy, although this could be argued as being implicit when fewer genes are shared in the community (i.e. lower niche overlap). We have now clarified this in the revised manuscript on lines 807-905.

L349: I am still not sure how you measured resource complexity in this study, and can therefore call this a driver of your results.

Answer: Resource complexity here is defined as the quality of the leaf litter, encapsulated by LDMC. This is in line with the excellent study of Garnier et al.²¹, showing that LDMC provides a “functional marker” of multiple ecosystem processes during secondary succession. We have now better clarified this in the discussion section (l. 759-760).

L358-366: Can you add these details earlier since they provide key justifications.

Answer: Absolutely! This section is rather crucial and we have now expanded it and put it early on in the discussion as the issues it addresses were of concern for all reviewers.

5. References

L581: Reference 6. This reference is incomplete and Arias-Navarro's name is misspelt.

Answer: This has been corrected

L756: Reference 84. Title needs spaces fixing.(Remarks on code availability)

Answer: This has been corrected

References

1. Bobay, L.-M. & Ochman, H. The Evolution of Bacterial Genome Architecture. *Front. Genet.* **8**, (2017).
2. Creamer, R. E. *et al.* Measuring basal soil respiration across Europe: Do incubation temperature and incubation period matter? *Ecol. Indic.* **36**, 409–418 (2014).
3. Manrubia, M. *et al.* Soil functional responses to drought under range-expanding and native plant communities. *Funct. Ecol.* **33**, 2402–2416 (2019).
4. Bradford, M. A. *et al.* Thermal adaptation of soil microbial respiration to elevated temperature. *Ecol. Lett.* **11**, 1316–1327 (2008).
5. Chapman, S. J., Campbell, C. D. & Artz, R. R. E. Assessing CLPPs using MicroResp™. *J. Soils Sediments* **7**, 406–410 (2007).
6. Eriksson, O., Cousins, S. A. O. & Bruun, H. H. Land-use history and fragmentation of traditionally managed grasslands in Scandinavia. *J. Veg. Sci.* **13**, 743–748 (2002).
7. Auffret, A. G., Kimberley, A., Plue, J. & Waldén, E. Super-regional land-use change and effects on the grassland specialist flora. *Nat. Commun.* **9**, 3464 (2018).
8. Mason, W. L. Changes in the management of British forests between 1945 and 2000 and possible future trends. *Ibis* **149**, 41–52 (2007).
9. Hooftman, D. A. P. & Bullock, J. M. Mapping to inform conservation: A case study of changes in semi-natural habitats and their connectivity over 70 years. *Biol. Conserv.* **145**, 30–38 (2012).
10. Mietkiewicz, N., Kulakowski, D., Rogan, J. & Bebi, P. Long-term change in sub-alpine forest cover, tree line and species composition in the Swiss Alps. *J. Veg. Sci.* **28**, 951–964 (2017).
11. Kuuluvainen, T., Tahvonen, O. & Aakala, T. Even-Aged and Uneven-Aged Forest Management in Boreal Fennoscandia: A Review. *AMBIO* **41**, 720–737 (2012).
12. Philippot, L., Chenu, C., Kappler, A., Rillig, M. C. & Fierer, N. The interplay between microbial communities and soil properties. *Nat. Rev. Microbiol.* **22**, 226–239 (2024).
13. Li, S. *et al.* Convergence and divergence in a long-term old-field succession: the importance of spatial scale and species abundance. *Ecol. Lett.* **19**, 1101–1109 (2016).
14. Tatoni, T. & Roche, P. Comparison of old-field and forest revegetation dynamics in Provence. *J. Veg. Sci.* **5**, 295–302 (1994).
15. Cramer, V. A., Hobbs, R. J. & Standish, R. J. What's new about old fields? Land abandonment and ecosystem assembly. *Trends Ecol. Evol.* **23**, 104–112 (2008).
16. Degens, B. P. & Harris, J. A. Development of a physiological approach to measuring the catabolic diversity of soil microbial communities. *Soil Biol. Biochem.* **29**, 1309–1320 (1997).
17. Moscatelli, M. C. *et al.* Assessment of soil microbial functional diversity: land use and soil properties affect CLPP-MicroResp and enzymes responses. *Pedobiologia* **66**, 36–42 (2018).
18. Bongiorno, G. *et al.* Soil management intensity shifts microbial catabolic profiles across a range of European long-term field experiments. *Appl. Soil Ecol.* **154**, 103596 (2020).
19. Mysara, M. *et al.* Reconciliation between operational taxonomic units and species boundaries. *FEMS Microbiol. Ecol.* **93**, fix029 (2017).
20. Cooke, R. S. C., Bates, A. E. & Eigenbrod, F. Global trade-offs of functional redundancy and functional dispersion for birds and mammals. *Glob. Ecol. Biogeogr.* **28**, 484–495 (2019).
21. Garnier, E. *et al.* Plant Functional Markers Capture Ecosystem Properties During Secondary Succession. *Ecology* **85**, 2630–2637 (2004).

Reviewer comments

Reviewer #1 (Remarks to the Author):

This study found that in the process of grassland to forest succession, soil microorganisms shifted from high taxonomic diversity with functional redundancy to lower taxonomic diversity with functional specialization. Forest succession enhanced the specialization of microbial C-N-P cycling genes while decreasing genetic redundancy. This finding provides insights into the functional changes in soil microbial communities following land abandonment. Overall, the study is logically rigorous and well-supported by the evidence. Moreover, all the comments from the reviewers and me were fully considered, so I have no more comments in this round. I think it is an interesting and excellent manuscript. I recommend it to be published on Nature Communications.

Answer: We thank the reviewer for their positive and constructive feedback throughout the revision process.

Reviewer #2 (Remarks to the Author):

This is a very well revised manuscript, and I applaud the co-authors for so carefully addressing each point raised during the initial review. The resulting manuscript has more depth, nuance, and it still captures its original and important point while placing the results into a richer context. The discussion is really much better and very interesting. Incorporating additional analyses to further explore your points (and including additional data on stand age and evenness) demonstrate a great effort and dedication. Well done!

Answer: We greatly appreciate the reviewer's feedback and are very pleased that our revisions have improved the manuscript.

I appreciate the additional discussion of the soil sampling x depth issue I raised in the methods section. While it remains an outlying "problem", and I am not convinced it doesn't affect interpretation of the results, I also understand that the authors feel this best aligns with their study goals. I admit, I am not so convinced because comparing soil with and without organic horizons and with varying levels of organic horizon material will shape every soil analysis. I nevertheless do not think nor suggest that this should prevent publications. If it were my paper, I would feel it necessary to add a statement in the discussion squarely saying something around the lines of "by sampling at a standard soil depth, we compare samples with and without organic horizons and with differing organic horizon depths, which strongly shapes resource availability, mineralogy, and physicochemical conditions that would be better captured by sampling at a consistent 0-10 cm depth in the mineral horizon". This would go well somewhere around L952. I leave it to the editor to decide whether this important or not.

Answer: We agree that sampling only the mineral horizon of the soils would make a better comparison in the aspects mentioned by the reviewer. However, this would exclude much of the microbial communities residing in the organic horizon and thereby miss the point of resource-driven community changes during succession. While we see no perfect solution to circumvent this problem, separate analyses on the organic and mineral horizons could perhaps offer additional insights at the cost of doubling the number of samples. We leave this for future studies to elucidate. We sincerely appreciate the suggested sentence and have

included a modified version of it into the method section (l. 470-476), which we feel reflect our view of this issue.

I also appreciate that the authors toned down causality (in many places), and discuss their results extremely effectively (great job with the revised work at L743-815). Nevertheless, there is still a considerable amount of causal language in the revised text. I would personally feel more comfortable, if this were my paper, to try and reduce this even further (e.g., at L44, 264, 265, 484, 486, etc.). Again, I leave this to the editor to design if it warrants additional attention.

Answer: We have gone through the manuscript again to tone down the causality further, especially at the suggested places (l. 35, 225, 226).

Good job on the revision. I look forward to seeing your work published.

Answer: Thank you!

Reviewer #3 (Remarks to the Author):
